# Genetic targeting of neurogenic precursors in the adult forebrain ventricular epithelium

Sandra E Joppé[1,2,*], Loïc M Cochard[1,3,*], Louis-Charles Levros, Jr[1,3], Laura K Hamilton[1,3], Pierre Ameslon[1,3], Anne Aumont[1], Fanie Barnabé-Heider[1], Karl JL Fernandes[1,3]

The ventricular epithelium of the adult forebrain is a heterogeneous cell population that is a source of both quiescent and activated neural stem cells (qNSCs and aNSCs, respectively). We genetically targeted a subset of ventricle-contacting, glial fibrillary acidic protein (GFAP)-expressing cells, to study their involvement in qNSC/aNSC–mediated adult neurogenesis. Ventricle-contacting GFAP+ cells were lineage-traced beginning in early adulthood using adult brain electroporation and produced small numbers of olfactory bulb neuroblasts until at least 21 mo of age. Notably, electroporated GFAP+ neurogenic precursors were distinct from both qNSCs and aNSCs: they did not give rise to neurosphere-forming aNSCs in vivo or after extended passaging in vitro and they were not recruited during niche regeneration. GFAP+ cells with these properties included a FoxJ1+GFAP+ subset, as they were also present in an inducible FoxJ1 transgenic lineage-tracing model. Transiently overexpressing Mash1 increased the neurogenic output of electroporated GFAP+ cells in vivo, identifying them as a potentially recruitable population. We propose that the qNSC/aNSC lineage of the adult forebrain coexists with a distinct, minimally expanding subset of GFAP+ neurogenic precursors.

## Introduction

The ventricular–subventricular zone (V-SVZ) surrounding the lateral ventricles is the largest germinal zone in the adult rodent brain, producing thousands of neuroblasts each day. V-SVZ neurogenesis derives from glial fibrillary acidic protein (GFAP)–expressing astrocytes (Doetsch et al, 1999a; Imura et al, 2003; Morshead et al, 2003; Garcia et al, 2004), a cell population that is scattered across both the ventricular zone (VZ) and subventricular zone (SVZ) compartments of the V-SVZ niche. The VZ compartment is a ciliated epithelium containing mainly ependymal cells and GFAP+ B1 astrocytes (Doetsch et al, 1997; Mirzadeh et al, 2008; Shen et al, 2008),

cells derived from a common embryonic precursor (Ortiz-Alvarez et al, 2019; Redmond et al, 2019) and that are intimately associated within pinwheel structures at the ventricular surface (Mirzadeh et al, 2008). Underlying the VZ is the SVZ compartment, which contains morphologically distinct subtypes of GFAP+ astrocytes, proliferating progenitors, migratory neuroblasts, and vasculature-associated cells (Doetsch et al, 1997; Mirzadeh et al, 2008; Shen et al, 2008; Tavazoie et al, 2008). GFAP+ cells in the VZ compartment are of particular therapeutic interest, as the ventricle-contacting population of GFAP+ B1 astrocytes includes cells having the properties of neural stem cells (NSCs) (Codega et al, 2014; Llorens-Bobadilla et al, 2015; Dulken et al, 2017). In clinical settings, these GFAP+ NSCs in the VZ can potentially be manipulated via the circulating cerebrospinal fluid.

Multiple types and/or stages of GFAP+ cells can be distinguished in the VZ compartment (Fig 1A and B). Within the population of GFAP+ B1 astrocytes are subsets of activated and quiescent NSCs (aNSCs and qNSCs, respectively). aNSCs are cycling, express the EGF receptor, and include the colony-forming neurosphere activity of the VZ. aNSCs in vivo appear to have a limited capacity for self-renewal (Calzolari et al, 2015; Obernier et al, 2018). Conversely, qNSCs are not cycling, EGF receptor-negative, and have a markedly delayed neurosphere-forming capacity (Codega et al, 2014; Llorens-Bobadilla et al, 2015; Dulken et al, 2017). Notably, the ability of sorted qNSCs to eventually give rise to neurosphere-forming aNSCs in vitro (Codega et al, 2014) suggests that aNSCs and qNSCs represent stages of a single neurogenic lineage (Codega et al, 2014; Chaker et al, 2016; Lim & Alvarez-Buylla, 2016; Obernier et al, 2018). Besides the GFAP+ B1 astrocyte population, the VZ also contains lesser studied subsets of GFAP+ cells that are integrated within the ependymal layer, such as transitional B1/ependymal cells (Luo et al, 2008), E2 ependymal cells (Mirzadeh et al, 2017), and niche astrocytes. The in vivo significance of these non–B1 GFAP+ cells is less understood.

In the present study, we sought to investigate the in vivo neurogenic properties of GFAP+ VZ cells, focusing specifically on their contributions to the aNSC pool and aNSC-mediated adult neurogenesis. In this regard, transgenic strategies have typically labeled

[1]Research Center of the University of Montreal Hospital (CRCHUM), Montreal, Canada    [2]Department of Pathology and Cell Biology, Faculty of Medicine, University of Montreal, Montreal, Canada    [3]Department of Neurosciences, Faculty of Medicine, University of Montreal, Montreal, Canada

Correspondence: karl.jl.fernandes@umontreal.ca
*Sandra E Joppé and Loïc M Cochard contributed equally to this work

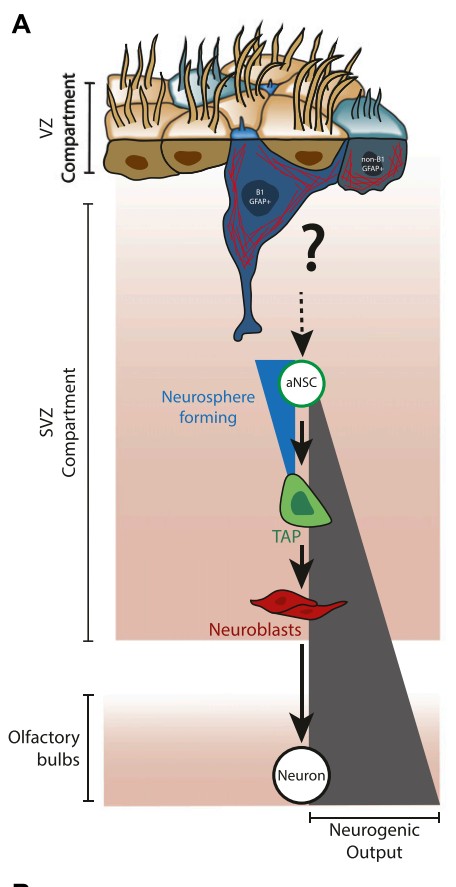

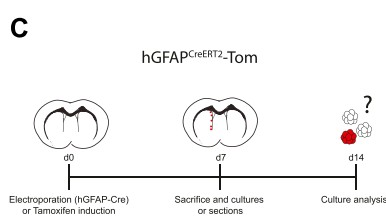

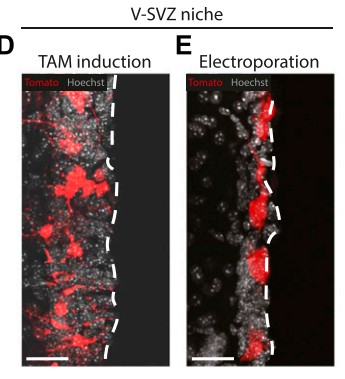

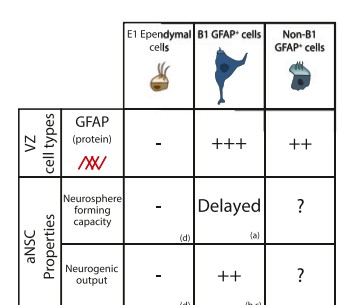

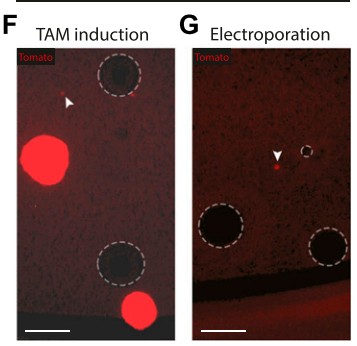

**Figure 1. Adult brain electroporation as an approach for studying the relationship of ventricle-contacting ventricular zone (VZ) cells and the activated neural stem cell population.**
**(A)** Anatomical organization and potential relationships between ventricle-contacting ependymal cells, B1 GFAP[+] cells, and non–B1 GFAP[+] cells (VZ compartment) and neurosphere-forming neural stem cells (SVZ compartment). **(B)** Table comparing key characteristics of these VZ cell types. **(C, D, E, F, G)** Electroporation to target ventricle contacting cells. **(C)** Experimental paradigm using hGFAP[CreERT2]-Tom mice. **(D, E)** Representative micrograph of Tomato[+] cells following tamoxifen induction (D) or electroporation of hGFAP-driven Cre plasmid (E). Note that electroporated cells are only located adjacent to the ventricular surface. **(F, G)** Representative micrograph of ventricular (V)-SVZ neurosphere cultures 1 wk after tamoxifen induction (F) or electroporation of hGFAP-Cre plasmid (G). Both conditions contain small Fluorescent colonies (arrowheads) but full-sized fluorescent neurospheres are present only in cultures from the tamoxifen-injected mice. Circles outline non-fluorescent neurospheres. References: (a) Codega (2014), (b) Mirzadeh (2008), (c) Obernier (2018), (d) Shah (2018). **(D, E, F, G)** Scale bars represent 30 μm in (D, E) and 100 μm in (F, G).

GFAP[+] cells across both the VZ and SVZ compartments (Mich et al, 2014), whereas division-dependent retroviruses infect the rare B1 cells that are cycling but not the vast majority of quiescent GFAP[+] cells (Doetsch et al, 1999a; Ihrie & Alvarez-Buylla, 2008; Obernier et al, 2018). Here, we used two separate strategies to investigate the neurogenic activity that originates from quiescent GFAP[+] cells located specifically in the VZ compartment. The first strategy, an electroporation-based approach (Barnabe-Heider et al, 2008), uses intraventricular plasmid injections to target GFAP promoter–expressing VZ cells via their ventricular contact. The second strategy, an inducible transgenic approach driven by the FoxJ1 promoter, enables genetic recombination within cilia-bearing cells of the VZ: this targets ependymal cells, which are non-neurogenic (Shah et al, 2018) and a FoxJ1[+] subset of ventricle-contacting GFAP[+] cells (Jacquet et al, 2009; Beckervordersandforth et al, 2010). Because neither of these fate-mapping strategies restricts genetic

targeting to dividing cells, they have enabled us to study the in vivo biological properties of quiescent GFAP[+] VZ cells. Our findings provide evidence that a subset of ventricle-contacting GFAP[+] cells, including at least FoxJ1[+]GFAP[+] cells, produces olfactory neuroblasts via an atypical non-aNSC route.

# Results

## Adult brain electroporation as a tool to genetically target GFAP[+] cells in the VZ

We first sought a method that would enable us to restrict genetic targeting and lineage-tracing to GFAP-expressing cells of the VZ compartment. Adult brain electroporation is a strategy in which

expression vectors are stereotaxically injected into the lateral ventricle cerebrospinal fluid and then electroporated into the striatal wall, enabling in vivo transfection of cells based on their anatomical contact with the ventricles (Barnabe-Heider et al, 2008). To test whether adult brain electroporation would restrict recombination to cells in the VZ compartment, we used tamoxifen-inducible hGFAP$^{CreERT2}$ transgenic mice crossed with Rosa26-stop-Tomato reporter mice (the cross herein referred to as hGFAP$^{CreERT2}$-Tom mice). Recombination in hGFAP$^{CreERT2}$-Tom mice was induced by either tamoxifen treatment (for global recombination in GFAP$^+$ cells) or by electroporation of hGFAP-driven Cre-recombinase plasmid (hGFAP-Cre; for local recombination in the VZ) (Fig 1C). After 7 d, tamoxifen-treated mice had recombined Tom$^+$ cells throughout the V-SVZ niche, as expected (Fig 1D). In contrast, hGFAP-Cre-electroporated mice had recombined cells only adjacent to the ventricular surface (Fig 1E). In mice processed for neurosphere cultures, recombined aNSC-associated neurospheres were observed in the tamoxifen-treated mice as previously reported (Mich et al, 2014) but not in hGFAP-Cre electroporated mice (Fig 1F and G). Given that the electroporated niche remains functional in terms of neurogenic output and yield of neurospheres (Fig S1A–E) (Barnabe-Heider et al, 2008), this suggested that aNSCs would not be directly transfected by the adult brain electroporation protocol.

Immunostaining was used to assess the cell types targeted by electroporation. When using fluorescent reporter plasmids driven by non–cell-specific promoters, an average of 1,174 ± 151 cells were labeled across the striatal VZ at 3 d post-electroporation (n = 5) (Fig 2A and B). At this early 3-d time point, we never observed co-staining with markers of non–ventricle-contacting cells such as SVZ transit-amplifying progenitors (Mash1$^+$ or Olig2$^+$) or neuroblasts (DCX$^+$) (Fig 2C). By using plasmids driven by the hGFAP promoter, reporter expression could then be enriched for GFAP-expressing VZ cells. For example, when mice were electroporated with a mixture of CMV-red fluorescent protein (RFP) and hGFAP-myrGFP (myristo-lated GFP) plasmids, hGFAP-myrGFP expression was restricted to 41% of all electroporated RFP$^+$ cells (Fig 2D). Immunofluorescence analysis of lateral ventricle whole-mount preparations further confirmed that the proportion of electroporated cells that were clearly positive for high levels of GFAP protein was increased five to sixfold when using plasmids driven from the hGFAP promoter than from a non–cell-specific promoter (Fig 2E and F). When we electroporated hGFAP-myrTom (myristolated Tomato) plasmids into the VZ of hGFAP::GFP transgenic mice, 93% of the Tom$^+$ cells were indeed GFP$^+$ (Fig 2G). Collectively, these data suggest that the vast majority of the 500–600 cells expressing hGFAP-driven plasmids per electroporated ventricle are indeed GFAP-expressing VZ cells.

The population of GFAP$^+$ VZ cells includes both B1 astrocytes (whose cell bodies are below the ependymal layer and extend a process to the ventricular surface) (Mirzadeh et al, 2008; Kokovay et al, 2012) and subsets of non–B1 GFAP$^+$ cells (whose cell bodies are intercalated within the ependymal layer itself) (Luo et al, 2008; Mirzadeh et al, 2017; Habela et al, 2020). Consistent with this, whole-mount analysis of the ventricular walls of mice electroporated with hGFAP-myrTom plasmids revealed two distinct morphologies: 9.3% ± 1.7% of Tom+ cells had the small, subependymal cell body with one or more branches characteristic of B1 cells (Fig 2H, top), and 90.7% ± 1.7% had larger cell bodies located directly at the ventricular

surface (Fig 2H, bottom) (6–7 fields from each of two whole-mounts). Immunostaining for GFAP protein on whole-mounts from naive control mice confirmed that GFAP$^+$ cells having these morphologies are likewise present in the non-electroporated VZ (Fig 2I).

Thus, using adult brain electroporation, we target 500–600 GFAP$^+$ VZ cells per electroporated ventricle, with the majority being non–B1 GFAP$^+$ cells.

## GFAP$^+$ VZ cells fate-mapped by adult electroporation produce neurons in vivo

Ventricle-contacting GFAP$^+$ cells were lineage-traced by electro-porating hGFAP-Cre plasmids into the VZ of Rosa26-stop-EYFP reporter mice (Fig 3A). When hGFAP-Cre was co-electroporated with hGFAP-myrTom plasmids, virtually 100% of Tom-expressing cells exhibited Cre-induced recombination of YFP expression by 7 d post-electroporation (Fig 3B), confirming this is an effective ap-proach to lineage-trace hGFAP plasmid-expressing cells. Initial analyses at 1 mo post-electroporation revealed the presence of rare recombined cells that were dividing (YFP$^+$EdU$^+$) or expressing neuroblast markers (YFP$^+$DCX$^+$) (Fig 3C and D). We, therefore, electroporated a large cohort of 3-mo-old Rosa26-stop-EYFP mice with hGFAP-Cre plasmids, analyzing them at time points between 1 and 72 weeks post-electroporation (WPE) to obtain a detailed time-course of the appearance of recombined neuroblasts.

Total numbers of YFP$^+$ cells within the V-SVZ did not exhibit a statistically significant change between 1 and 21 WPE but ap-proximately doubled by 72 WPE (Fig 3E). YFP$^+$DCX$^+$ recombined neuroblasts were not detectable in the V-SVZ at 1 WPE. However, at 4 WPE and all time points up to 72 WPE, they represented 1–3% of the total number of recombined YFP$^+$ cells (Fig 3F, y1-axis). The estimated absolute number of recombined neuroblasts per V-SVZ was highest at 72 WPE (Fig 3F, y2-axis), and the overall proportion of mice having recombined neuroblasts showed a time-dependent increase: YFP$^+$DCX$^+$ neuroblasts were detected in 55% of animals at 4–16 WPE (10/18 mice) and in 90% of mice at 21–72 WPE (9/10 mice). It was noted that 2 of the 31 animals in this time-course experiment had highly elevated numbers of recombined cells (1/4 mice at 4 WPE and 1/5 mice at 12 WPE): more detailed analysis of these two animals revealed no clusters or differences in spatial localization of recom-bined cells, suggesting that the electroporation initially recombined more cells rather than in a different cell population (Fig S2).

Examination of the olfactory bulbs (OBs), the typical destination of V-SVZ neuroblasts, indeed revealed small numbers of highly differentiated YFP$^+$ neurons (Fig 3G). Interestingly, quantification showed that the number of recombined OB neurons did not ex-hibit the same time-dependent increase as observed for YFP$^+$DCX$^+$ neuroblasts and total YFP$^+$ cells in the V-SVZ, perhaps reflecting age-related changes in neuroblast migration, survival, or integra-tion. Recombined neurons were present in the OBs of 2/4 mice at 4 WPE, 5/5 mice after 12 wk, 3/3 mice after 21 wk, and 5/5 mice after 72 wk (Fig 3H).

Thus, when a single cohort of 500-600 GFAP$^+$ VZ cells is lineage-traced beginning at 3-mo of age, the vast majority of electroporated adult mice will eventually generate YFP$^+$DCX$^+$ recombined neuro-blasts and will continue producing such neuroblasts until at least 21 mo of age.

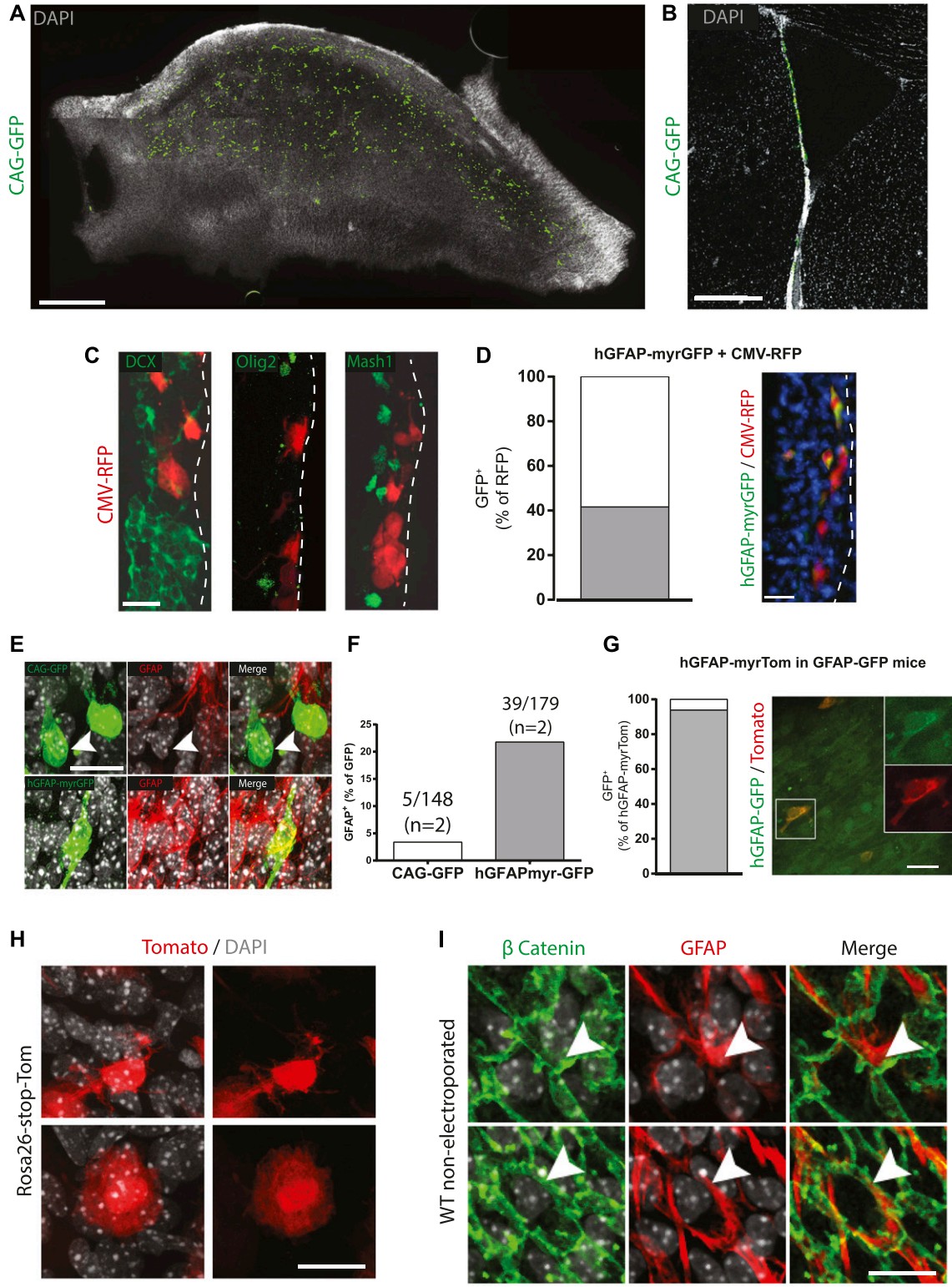

**Figure 2.   Characteristics of ventricle-contacting ventricular zone cells targeted by adult brain electroporation.**
**(A, B, C)** Adult brain electroporation using plasmids driven by non–cell-specific CMV regulatory elements. **(A, B)** Distribution of electroporated cells within the ventricular–subventricular zone are shown on a (A) whole-mount and (B) coronal section. **(C)** Immunostaining of electroporated cells after 3 d showing no co-localization with sub-ependymal DCX-, Olig2-, or Mash1-positive cells (n = 5). **(D, E, F, G, H, I)** Electroporation using reporter plasmids driven by the hGFAP promoter. **(D)** Co-electroporation of hGFAP-myrGFP with CMV-RFP plasmids. Quantification of the RFP⁺ cells expressing GFP and micrograph of a coronal section. **(E)** Immunostaining of WT mice electroporated with CAG-GFP (GFAP negative, arrowhead, upper panels) or hGFAP-myrGFP (GFAP positive, lower panels) plasmids in WT mice. **(F)** Quantification of

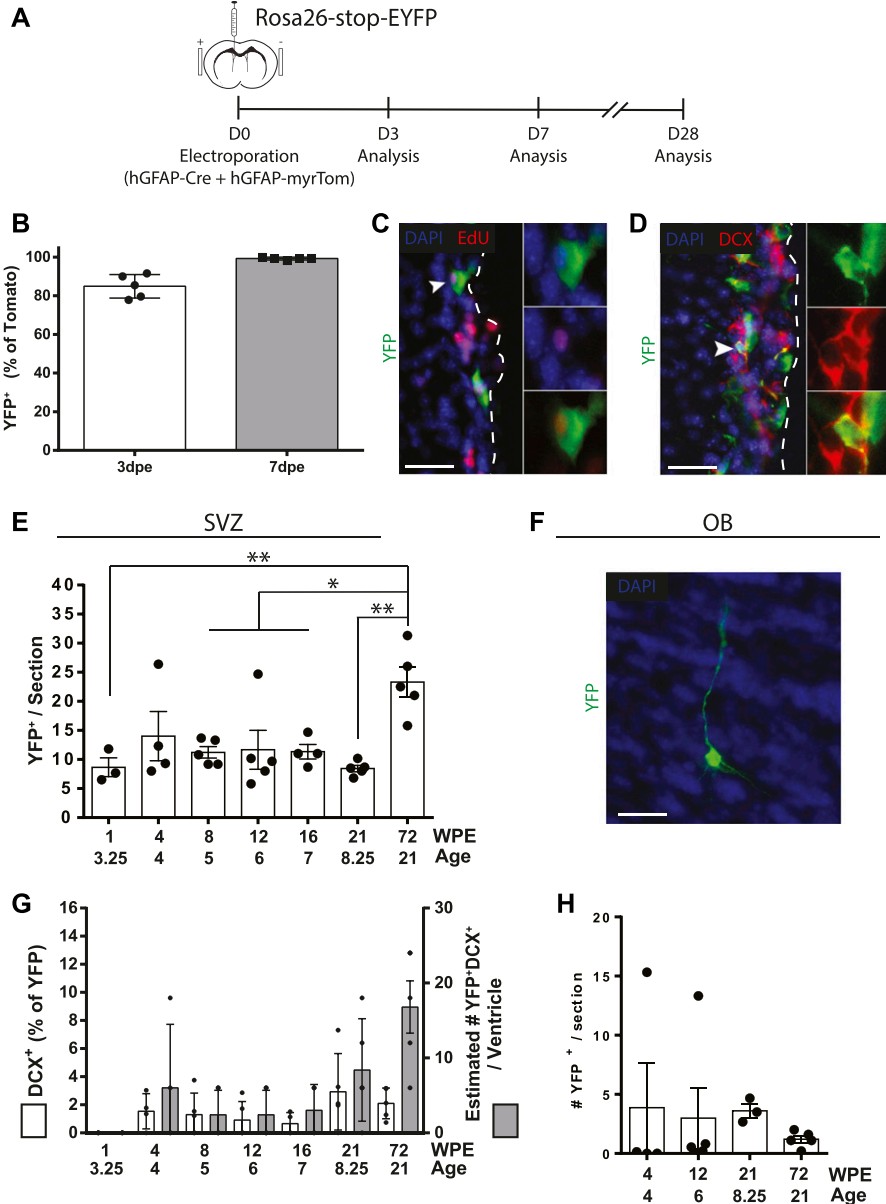

**Figure 3. Neurogenic properties of ventricle-contacting GFAP+ cells genetically targeted by adult brain electroporation.**

**(A, B, C, D)** hGFAP-Cre–mediated recombination model. **(A)** Paradigm. 3-mo-old Rosa26-stop-EYFP reporter mice were electroporated with 5 µg hGFAP-myrTom and 5 µg hGFAP-Cre plasmids and analyzed after 3 or 7 d. **(B)** Quantification of the proportion of YFP+ cells that expressed Tom. Note that Cre-induced recombination of YFP expression occurred in the vast majority of Tom+ cells by 3 d and virtually all Tom+ cells by 7 d. **(C, D)** After analysis at 28 d post-electroporation, representative micrographs of recombined cells that incorporated EdU (arrowhead) (C) or expressed the neuroblast marker DCX (arrowhead) (D) in the ventricular–subventricular zone (V-SVZ). **(E, F, G, H)** Time-course of neurogenesis by recombined cells up to 72 weeks post-electroporation of hGFAP-Cre plasmids in 3-mo-old ROSA-stop-EYFP reporter mice. **(E)** YFP+ cells after 1 (n = 3), 4 (n = 4), 8 (n = 5), 12 (n = 5), 16 (n = 4), 21 (n = 5), or 72 (n = 5) weeks post-electroporation. **(F)** YFP+DCX+ neuroblasts in terms of percentage of YFP+ cells (y1-axis) and estimated total numbers (y2-axis). **(G, H)** In the OB, (G) micrograph and (H) quantification of YFP+ cells at the 4-, 12-, 21-, and 72-wk time points. Age, age at sacrifice (months); WPE, weeks post-electroporation. One-way ANOVA, Tukey's multiple comparisons test. **(C, D, G)** Scale bars represent 30 µm in (C, D, G). *$P \leq 0.05$ **$P \leq 0.01$.

## GFAP+ VZ cells fate-mapped by adult electroporation do not give rise to aNSCs

A hallmark of aNSCs and their immediate transit-amplifying progeny is the ability to generate neurosphere colonies in response to EGF (Morshead et al, 1994; Doetsch et al, 2002; Codega et al, 2014). Previous studies have shown that V-SVZ neurogenesis is mediated by neurosphere-forming aNSCs and that aNSCs can be produced by a population of quiescent, ventricle-contacting GFAP+ NSCs (Imura et al, 2003; Morshead et al, 2003; Garcia et al, 2004; Codega et al, 2014; Mich et al, 2014). We, therefore, asked whether the neurogenic, GFAP+ VZ cells labeled by adult brain electroporation exhibit the neurosphere-forming characteristics of aNSCs and/or qNSCs (Fig 4A).

GFP+ cells unambiguously expressing high levels GFAP protein. **(G)** Electroporation of hGFAP-myrTom plasmids in hGFAP::GFP transgenic mice. Quantification of Tomato+ cells expressing GFP and micrograph of a whole-mount. **(H)** Representative images of the two morphologies of cells electroporated with hGFAP-driven plasmid: B1-like cells with small cell bodies and extensions toward the ventricular surface (upper panels) and non-B1 cells having larger cell bodies located directly at the ventricular surface (lower panels). **(I)** Micrographs of a ventricular–subventricular zone whole-mount from a non-electroporated animal, stained with β catenin and GFAP, also showing the presence of B1 and non–B1 GFAP+ cells (arrowheads in the upper and lower panels, respectively). **(A, B, C, D, E, G, H, I)** Scale bars represent 500 µm in (A), 250 µm in (B), 25 µm in (C, D, E), 20 µm in (G) and 15 µm in (H, I).

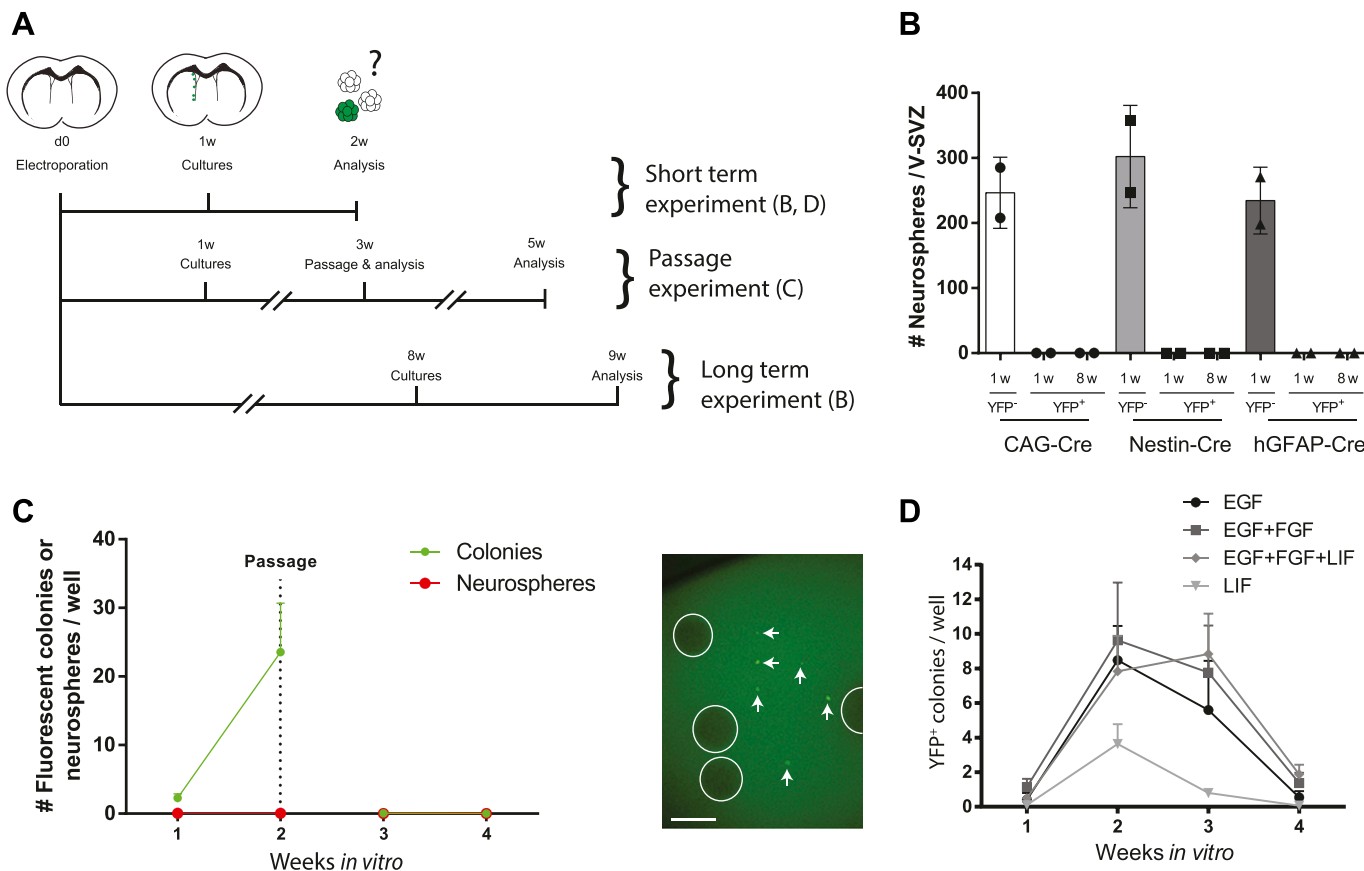

**Figure 4. Electroporated ventricular zone (VZ) cells do not produce neurosphere-forming activated neural stem cells in vitro or in vivo.**
**(A)** Paradigms. 3-mo-old ROSA-stop-EYFP reporter mice were electroporated with Cre-expressing plasmids and traced in vivo for short-term or long-term periods (1 or 8 wk, respectively) before conducting neurosphere-forming assays. **(B)** No recombined neurospheres were generated, regardless of whether lineage-traced in vivo for 1- or 8-wk and electroporated with Nestin-Cre, hGFAP-Cre, or CAG-Cre plasmids. **(C)** For the passaging experiment, ventricular–subventricular zone (V-SVZ) cultures were generated 1-wk after electroporation with hGFAP-Cre plasmid, re-fed with EGF-containing condition, and passaged after 2 wk in vitro. Fluorescent colonies were lost upon passaging, and fluorescent neurospheres were not generated. **(D)** Cells that recombined after hGFAP-Cre electroporation only formed small YFP⁺ colonies (left, arrows), which disappeared over time, regardless of growth factor combination used (right) (n = 3 per condition). Circles outline non-fluorescent neurospheres. **(D)** Scale bars represent 100 μm in (D).

We first electroporated 3-mo-old Rosa26-stop-EYFP reporter mice with hGFAP-Cre plasmids as previously and after 1-wk the electroporated V-SVZs were dissociated and grown in standard EGF-containing neurosphere conditions. Cultures from electroporated V-SVZs produced normal total numbers of neurospheres, but these cultures were completely devoid of YFP⁺ neurospheres (Fig 4B); instead, YFP⁺ cells only formed small clusters that failed to grow beyond 5–10 cells even if continuously fed for 1 mo. Identical results were obtained when we repeated this experiment using Nestin-Cre (expressed in aNSCs/early transit-amplifying progenitors) or CAG-Cre (non-cell-specific) plasmids (Fig 4B). These data stand in contrast to the earlier results using tamoxifen-treated GFAP^CreERT2 mice (Fig 1D) and indicate adult brain electroporation labels a subset of neurogenic, GFAP⁺ VZ cells that are distinct from neurosphere-forming aNSCs.

We then asked whether electroporated GFAP⁺ VZ cells would give rise to neurosphere-forming aNSCs if provided with additional time or additional growth factor support, as reported for qNSCs (Codega et al, 2014) or "primitive" NSCs (Reeve et al, 2017), respectively.

However, we found no evidence that this is the first case. First, when we extended the interval between electroporation and neurosphere culturing from 1 to 8 wk, YFP⁺ neurospheres were still not obtained regardless of whether Cre expression had been driven by the hGFAP, Nestin, or CAG promoters (Fig 4B). Second, when V-SVZs were electroporated and cultured after 1 wk, then re-fed weekly using standard EGF-containing conditions and passaged after 2 wk in vitro, fluorescent neurospheres were still not produced and the small fluorescent colonies disappeared (Fig 4C). Last, we repeated the above electroporation paradigm using hGFAP-Cre plasmids and then cultured the V-SVZs in either EGF alone, EGF+FGF2, EGF+FGF2+LIF, or LIF alone. Quantification showed that normal total numbers of neurospheres were generated in all EGF-containing conditions, and again, no YFP⁺ neurospheres were obtained in any condition (not shown). As previously, YFP⁺ cells only formed small clusters that were apparent at 7 d in vitro, peaked in number around 14 d in vitro, and virtually disappeared by 28 d in vitro, regardless of growth factor combination (Fig 4D, right). Thus, the neurogenic, GFAP⁺ VZ cells labeled by electroporation still fail to produce

neurosphere-forming aNSCs after prolonged in vivo lineage tracing, extended passaging, or treatment with additional NSC growth factors.

Together, these data show that despite having in vivo neurogenic capacity, electroporated GFAP$^+$ VZ cells are i) not themselves neurosphere-forming aNSCs and ii) unlikely to correspond to previously described upstream NSCs, such as qNSCs or LIF-responsive "primitive" NSCs.

### FoxJ1$^+$ cells in the VZ produce limited numbers of OB neurons

We next sought a mechanistically distinct approach for lineage-tracing cells in the VZ compartment to validate these findings. FoxJ1 is a transcription factor that directs expression of cilia-related genes in epithelial cells. In the V-SVZ niche, its expression is restricted to the VZ where it is expressed by all ependymal cells and a subpopulation of GFAP$^+$ cells (Jacquet et al, 2009; Beckervordersandforth et al, 2010; Codega et al, 2014). We immunostained coronal sections of the V-SVZ of wild-type mice and confirmed that FoxJ1 protein was indeed present within the nucleus of both ependymal cells and occasional GFAP$^+$ cells (Fig 5A). On lateral ventricle whole-mount preparations from GFAP::GFP transgenic reporter mice, FoxJ1 immunoreactivity within GFAP-GFP$^+$ cells was relatively lower than in the surrounding ependymal cells (Fig 5B). Notably, on whole-mounts from wild-type mice electroporated with hGFAP-myrGFP, an estimated 70% of the GFP$^+$ cells were immunoreactive for FoxJ1 protein, indicating that a major proportion of hGFAP electroporated cells are FoxJ1$^+$GFAP$^+$ (Fig 5C).

To determine whether cells within the FoxJ1-expressing population participate in V-SVZ neurogenesis, we took advantage of FoxJ1$^{CreERT2}$ knock-in mice in which tamoxifen-inducible CreERT2 is knocked into one allele of the endogenous FoxJ1 locus (Muthusamy et al, 2014). FoxJ1$^{CreERT2}$ knock-in mice were crossed with Rosa26-stop-EYFP mice (the cross herein referred to as FoxJ1$^{CreERT2}$-EYFP mice) to enable tamoxifen-induced YFP expression in FoxJ1$^+$ cells and their progeny. 4 wk after tamoxifen administration to 3-mo-old FoxJ1$^{CreERT2}$-EYFP mice, strong YFP labeling was present along the entire ependyma of the ventricular system, confirming robust recombination in the VZ (Fig 5D and I). Recombination was highly specific (99.43% ± 0.04% of YFP$^+$ cells indeed expressed FoxJ1 protein) and had an overall efficiency of more than 50% (52.1% ± 2.56% of all cells expressing FoxJ1 protein were YFP$^+$, Fig 5D and E). Immunostaining showed that 3.95% ± 0.16% of the FoxJ1-YFP$^+$ cells expressed GFAP protein (850-1152 YFP$^+$ cells analyzed/animal, n = 4 animals), and these FoxJ1$^+$GFAP$^+$ were equally distributed along the dorsoventral axis (Fig 5F–H). Analysis of the V-SVZ niche revealed small numbers of recombined YFP$^+$ cells that co-expressed the neuroblast marker DCX. YFP$^+$DCX$^+$ cells were present in 3/3 mice at 4 wk post-tamoxifen (18/1418 YFP$^+$ cells, n = 3) and 3/3 mice at 16 wk post-tamoxifen (11/905 YFP$^+$ cells, n = 3) (Fig 5J and K). Because DCX expression is limited to a short time-window after neurogenesis, this suggests that cells within the FoxJ1-expressing VZ population continue to produce new neurons until at least 7 mo of age. Consistent with this, examination of the OBs revealed the presence of rare YFP$^+$ cells in the granular zone that increased in number between 4 and 16 wk post-tamoxifen (4 wk, 0.7 ± 0.3 YFP$^+$ cells/section, n = 5; 16 wk, 1.8 ± 0.3 YFP$^+$ cells/section, n = 4) (Fig 5L and M).

Thus, the FoxJ1-expressing population in 3-mo-old mice represents a continual source of small numbers of new neurons until at least 7 mo of age.

### FoxJ1$^+$ VZ cells do not contribute significantly to the aNSC pool

To test for a lineage relationship between neurogenic cells in the FoxJ1$^+$ VZ population and neurosphere-forming aNSCs, young adult FoxJ1$^{CreERT2}$-EYFP mice were tamoxifen-treated and then processed for lateral ventricle-derived neurosphere cultures (Fig 6A). When neurosphere cultures were generated within 2 wk of tamoxifen treatment, cultures derived from the lateral ventricles each yielded hundreds of total neurospheres but were completely devoid of YFP$^+$ neurospheres (Fig 6B and C). As a positive control for these experiments, we also generated neurosphere cultures from the spinal cords of these mice, as neurosphere-forming cells in the spinal cord originate from within the FoxJ1-expressing ependymal cell population (Meletis et al, 2008; Barnabe-Heider et al, 2010). Consistent with the ~50% recombination rate in these mice, half of the neurospheres in spinal cord–derived cultures of these same mice were indeed YFP$^+$ (Fig 6C). Thus, neurogenic FoxJ1$^+$ cells in the forebrain VZ are not part of the neurosphere-forming NSC pool.

We then extended the in vivo lineage-tracing time (i.e., the interval between tamoxifen treatment and V-SVZ culturing) to 16 wk to determine whether FoxJ1$^+$ VZ cells might contribute to the aNSC pool over a longer timeframe. Hundreds of total neurospheres were again generated in the lateral ventricle cultures derived from each of four mice and, interestingly, a single YFP$^+$ neurosphere was now produced in cultures from three of four of these mice (1.3 ± 0.5 YFP$^+$ neurospheres/V-SVZ, n = 4) (Fig 6D).

Together, these lineage tracing studies reveal that the FoxJ1$^+$ population of VZ cells includes a subset of cells that are neurogenic in vivo. These neurogenic FoxJ1$^+$ cells i) are not neurosphere-forming aNSCs, ii) do not undergo spontaneous lineage progression into neurosphere-forming aNSCs upon removal from their niche, but iii) might have the capacity to sporadically contribute to the aNSC pool over the long term.

### Electroporated VZ cells do not contribute to niche regeneration

The above fate-mapping strategies using adult brain electroporation (hGFAP promoter-driven) and transgenic mice (FoxJ1 promoter-driven) both indicated that the ventricular epithelium contains a subpopulation of cells that produces small numbers of olfactory neurons yet shows no evidence of contributing significantly to the aNSC pool. To investigate whether this subset of cells might be recruited to play a role during niche regeneration and repair, we studied their responses in the well-characterized Ara-C model of niche depletion and repopulation (Doetsch et al, 1999a, 1999b). In this model, administration of the anti-mitotic agent cytosine-arabinoside (Ara-C) depletes the V-SVZ niche of its actively dividing aNSCs, progenitors, and neuroblasts, and upon arrest of Ara-C treatment, these populations are normally replenished by more qNSCs.

Rosa26-stop-EYFP reporter mice were electroporated with hGFAP-Cre plasmids as previously and then immediately implanted

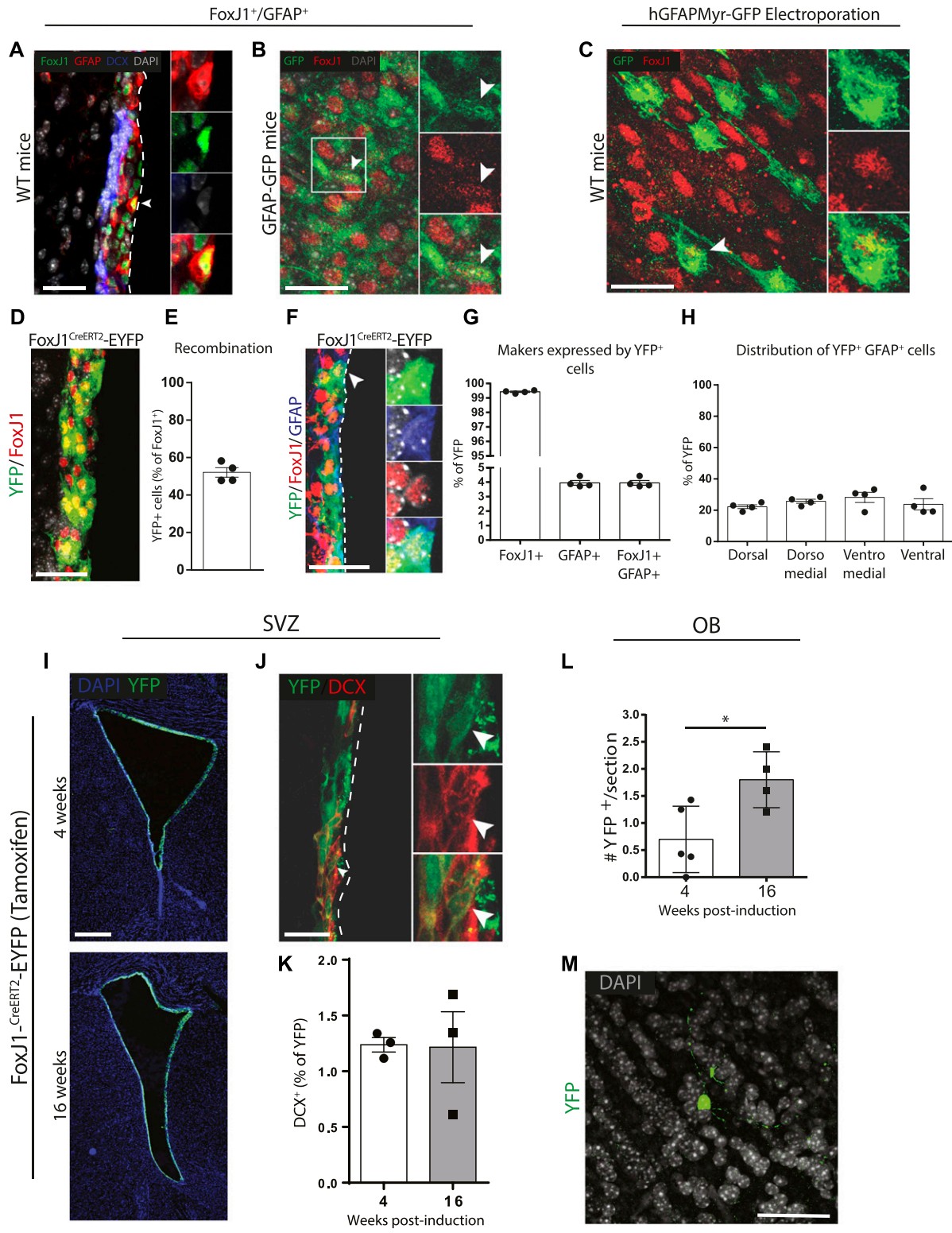

**Figure 5. The adult FoxJ1+ population contains neurogenic cells.**
**(A, B, C)** FoxJ1+GFAP+ cells are present in the ventricular zone (VZ). **(A)** In WT mice, FoxJ1 protein is detectable in some cells expressing GFAP protein (arrowhead). **(B)** In ventricular–subventricular zone (V-SVZ) whole-mounts of GFAP-GFP transgenic mice, a subpopulation of GFP+ cells expresses low levels of FoxJ1 protein (arrowhead). **(C)** After electroporation with GFAP-myrGFP, a subset of GFP+ cells at the ventricular surface of whole-mounts express FoxJ1 protein (arrowhead). **(D, E, F, G, H, I, J, K, L, M)** In vivo neurogenesis by FoxJ1+ cells. Young adult FoxJ1^CreERT2–EYFP compound transgenic mice were administered tamoxifen for 1 wk and euthanized after 4 or 16 wk (n = 3/group). **(D)** Representative micrograph showing YFP is recombined in a large portion of FoxJ1-expressing cells. **(E)** Quantification of recombination in the FoxJ1-

Short term

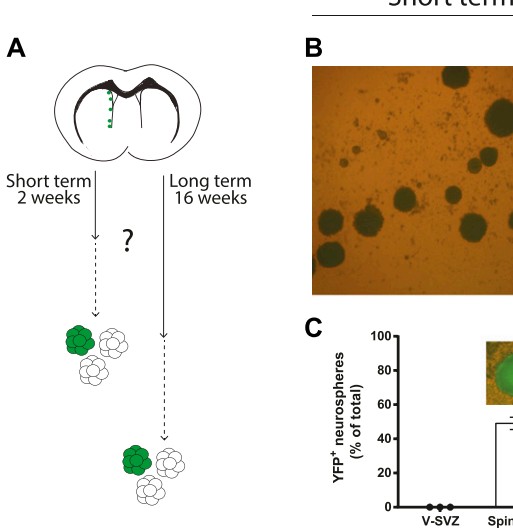

Long Term

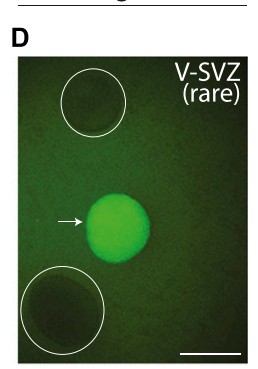

**Figure 6. Neurogenic FoxJ1+ cells are not neurosphere-forming activated neural stem cells.**
**(A, B, C, D)** FoxJ1+ cells in the ventricular–subventricular zone (V-SVZ) are not a significant source of neurosphere-forming neural stem cells. **(A)** Recombined cells in tamoxifen-treated FoxJ1^CreERT2_-EYFP mice were traced in vivo for short- or long-term periods before conducting neurosphere-forming assays, as indicated in (A). **(B, C)** In the short-term paradigm, (B) representative micrographs of neurospheres grown from the V-SVZ and spinal cord and (C) quantifications of the neurospheres that were YFP+. **(D)** In the long-term paradigm, (D) a micrograph of one of the extremely rare recombined neurospheres that was obtained from the V-SVZ after 4-mo of in vivo lineage-tracing (arrow). Circles outline non-fluorescent neurospheres. **(B, D)** Scale bars represent 100 μm in (B, D).

with intracerebroventricular (ICV) osmotic pumps containing Ara-C or vehicle (Fig 7A). As expected, 1 wk of Ara-C infusion drastically decreased total numbers of proliferating cells (Ki67+) and neuroblasts (DCX+) in the V-SVZ (Fig 7B–D). Analysis of the number of YFP+ cells revealed no significant changes at the end of 7 d of Ara-C treatment, indicating that recombined GFAP+ VZ cells remained dormant during the depletion of mitotic cells (Fig 7E). In animals that underwent an additional 21 d of V-SVZ repopulation, robust regeneration of the V-SVZ niche was observed as the Ki67 and DCX populations were found to be completely restored to normal levels (Fig 7F–I). In these repopulated animals, the total number of YFP+ cells (Fig 7J) and YFP+DCX+ neuroblasts (Fig 7K) in the V-SVZ did not expand in number; to the contrary, there was a tendency toward a decrease in total YFP+ cells (Fig 7J). Thus, electroporated GFAP+ cells in the VZ are not the dormant precursors that regenerate the V-SVZ niche and show no evidence of contributing to the recovery of proliferation and neurogenesis.

## Overexpression of Mash1 in ventricle-contacting GFAP+ cells

The proneurogenic transcription factor Mash1/Ascl1 is a critical regulatory point in the transition from NSC quiescence to activation (Imayoshi et al, 2013; Llorens-Bobadilla et al, 2015; Urban et al, 2016; Sueda et al, 2019). Mash1 is expressed by aNSCs and their immediate downstream progenitors, is degraded during their quiescence, and can be overexpressed in non-neural cell types to drive direct neuronal transdifferentiation (Berninger et al, 2007; Heinrich et al, 2012). We, therefore, asked whether we could use Mash1 to promote neuronal differentiation from the population of quiescent GFAP+ VZ cells.

As proof of concept, plasmids encoding Mash1 were tested using in vitro neurosphere cultures. Undifferentiated secondary neurosphere cells were transfected with RFP reporter plasmids that were mixed with either Mash1 or empty vector (EV) plasmids and then allowed to differentiate for 2 d (Fig 8A). Mash1 transfection increased the proportion of RFP+ cells expressing the neuronal marker betaIII-tubulin from 0.43% ± 0.43% to 21.21% ± 2.78% (P = 0.0018, n = 3), an ~40-fold increase (Fig 8B and C). Thus, Mash1 overexpression is sufficient to drive neural stem/progenitor cells to a neuronal fate in vitro.

We, therefore, performed fate-mapping in 3-mo-old Rosa26-stop-EYFP reporter mice electroporated with hGFAP-Cre plasmids mixed with either Mash1 versus EV plasmids (n = 10/group) (Fig 8D). Because analysis after 7 d suggested that Mash1 overexpression reduced recombination driven from the hGFAP promoter (Fig 8E), we also electroporated additional reporter mice with CAG-Cre plasmids mixed with Mash1 versus EV (n = 5/group) plasmids. Both hGFAP-Cre and CAG-Cre recombination models were then assessed for the impact of Mash1 overexpression.

Mash1 plasmids increased the proportion of electroporated cells expressing high levels of Mash1 immunoreactivity from less than 1% to 20–30% of recombined cells (Fig 8F). Double-labeling revealed that, in both recombination models, the total number of YFP+DCX+ neuroblasts remained relatively low upon Mash1 overexpression (Fig 8G). However, because recombined DCX+ neuroblasts are not normally present at 7 d post-electroporation, the increase in YFP+DCX+ neuroblasts in the hGFAP-Cre recombination model approached statistical significance (P = 0.054, t test versus EV) (Fig 8G). Despite its limited effect size, Mash1 overexpression increased the proportion of electroporated mice having YFP+DCX+ neuroblasts

expressing population. **(F)** Representative micrograph of an YFP+ cell that expresses GFAP and FoxJ1 proteins (arrowhead). **(G)** Quantification of GFAP and FoxJ1 expression in the YFP+ population. **(H)** Dorsoventral distribution of YFP+ cells that are FoxJ1+GFAP+. **(I, J, K, L, M)** Representative micrographs of (I) the recombined cells surrounding the lateral ventricles, (J) an YFP+DCX+ neuroblast (arrowhead), and (K) quantification of the YFP+DCX+ neuroblasts. **(L, M)** In the OB, (L) quantification of the numbers of YFP+ cells and (M) a representative micrograph. **(A, B, C, D, F, M, J, I)** Scale bars represent 25 μm in (A, B, C, D, F, M), 35 μm in (J), and 200 μm in (I). *P ≤ 0.05, unpaired t tests.

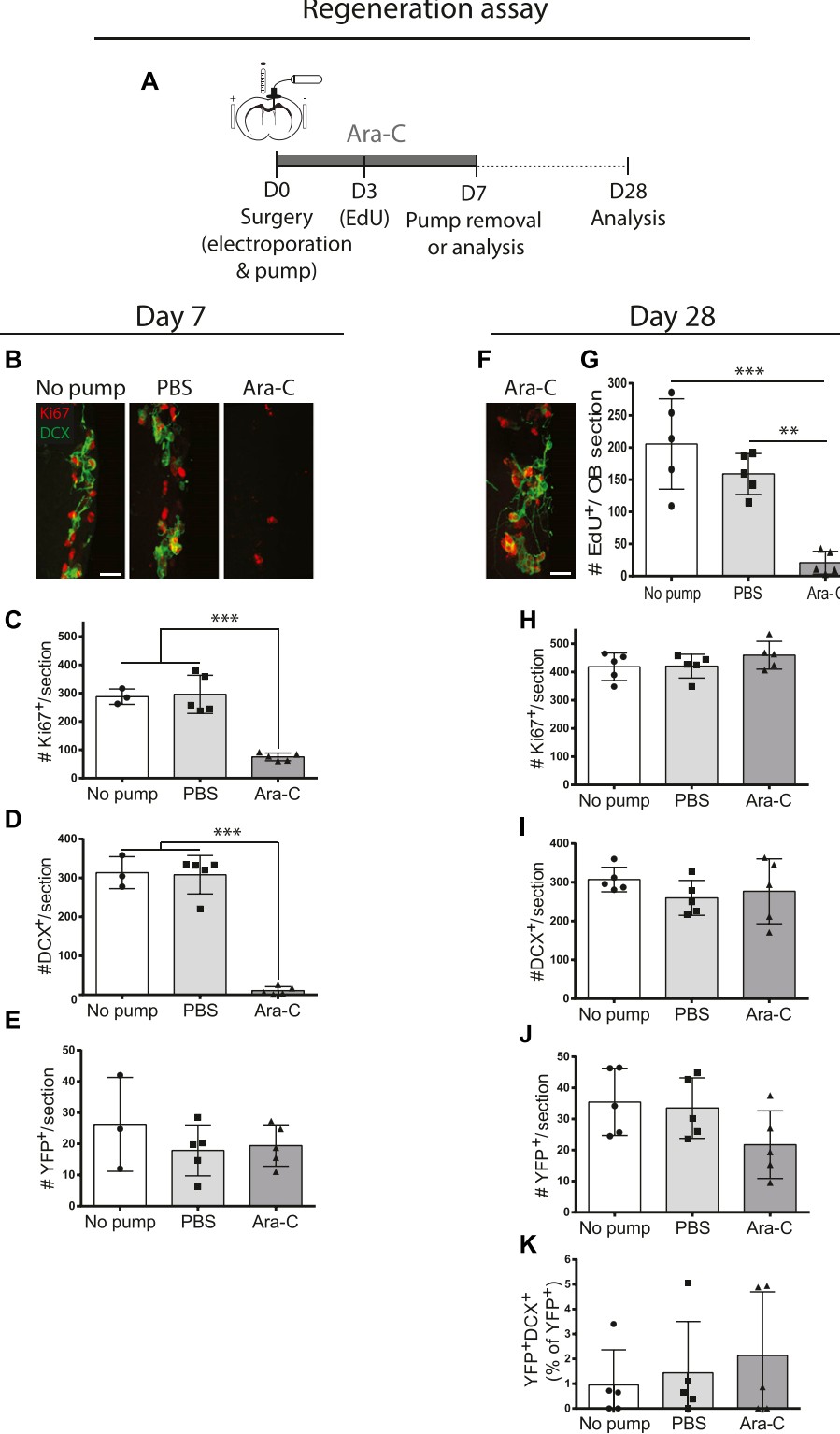

**Figure 7. hGFAP⁺ B1 cells are not recruited in the Ara-C model of ventricular–subventricular zone (V-SVZ) depletion and regeneration.**
**(A)** Ara-C experimental paradigm after electroporation of hGFAP-Cre plasmids in 3-mo-old ROSA-stop-EYFP reporter mice. **(B, C, D, E)** Analysis at Day 7. **(B)** Immunostaining for Ki67 and DCX in the V-SVZ of no pump, PBS, and Ara-C mice. **(C, D, E)** Quantification of numbers of V-SVZ cells that were immunoreactive for (C) Ki67, (D) DCX, or (E) YFP. One-way ANOVA. **(F, G, H, I, J, K)** Analysis at Day 28. **(F)** Immunostaining for Ki67 and DCX in the V-SVZ of Ara-C mice. **(G)** Numbers of EdU⁺ cells in the OB, confirming that V-SVZ neurogenesis had been ablated in Ara-C mice. **(H, I, J, K)** Quantification of numbers of V-SVZ cells that were immunoreactive for (H) Ki67, (I) DCX or (J) YFP, and (K) the proportion of YFP⁺ cells expressing DCX (n = 5 per time point and per condition). One-way ANOVA. **(B, F)** Scale bar represents 20 $\mu$m in (B) and (F). *$P \leq 0.05$ **$P \leq 0.01$, ***$P \leq 0.001$.

from 30% (3/10) with EV to 67% (10/15) with Mash1. Interestingly, Mash1 overexpression was not sufficient to stimulate neurosphere-forming competence in the electroporated GFAP⁺ cells (data not shown), but it often resulted in clusters of recombined neuroblasts in vivo, suggestive of clonal expansion (Fig 8H). Pooling of the electroporation data from the two recombination models showed that the proportion of YFP⁺DCX⁺ neuroblasts present at an early 7-d time point increased from 0.15% of recombined cells with

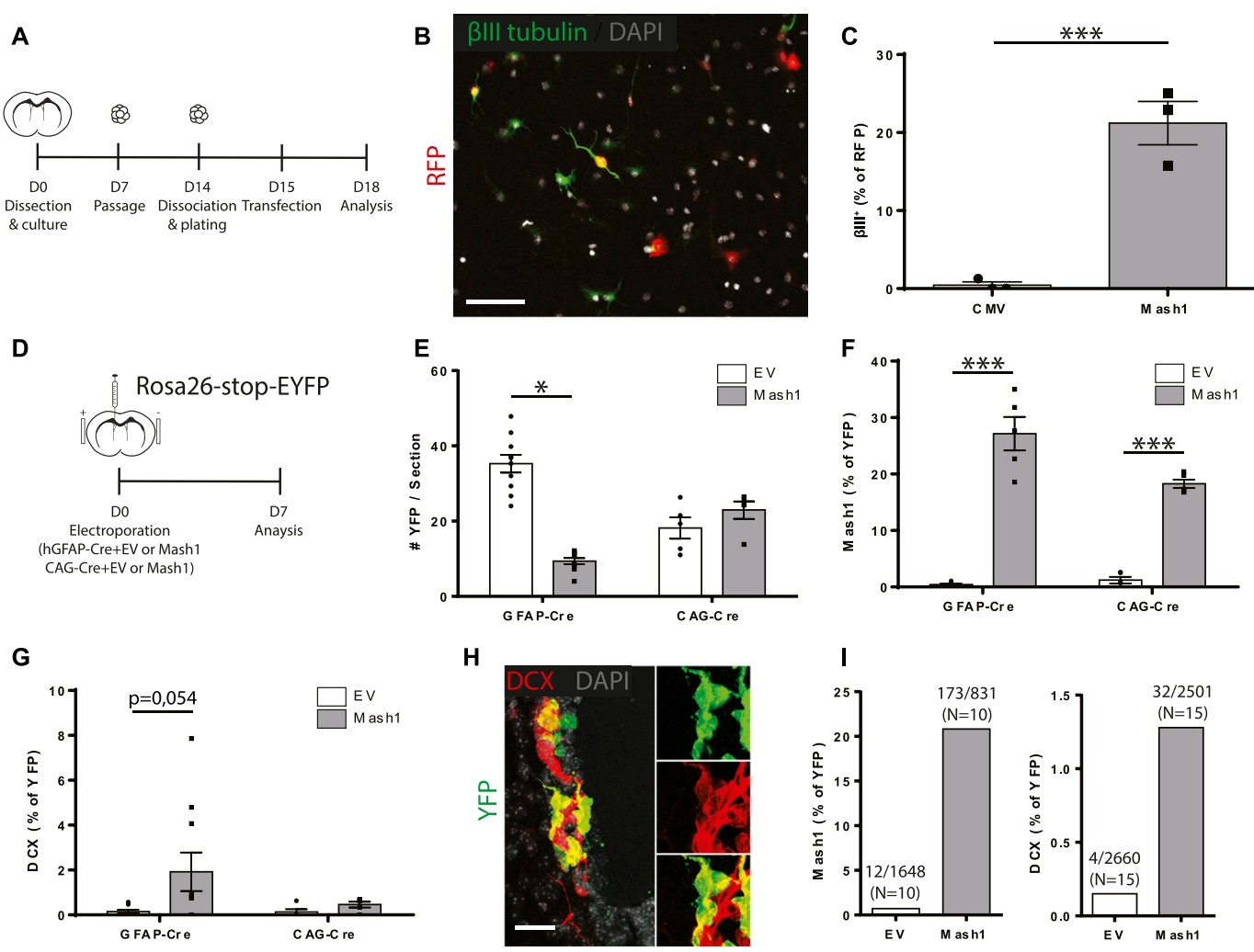

**Figure 8.  Overexpression of Mash1 in electroporated GFAP⁺ ventricular zone cells.**
**(A, B, C)** Co-transfection of neurosphere-derived stem/progenitor cells with RFP and empty vector or Mash1 plasmids in vitro. **(A)** Experimental paradigm. **(B, C)** Micrograph of transfected RFP⁺ cells expressing βIII tubulin and (C) quantification of RFP⁺ cells expressing βIII tubulin. Unpaired *t* test (n = 3/condition). **(D, E, F, G, H, I, J)** Co-electroporation of ventricular zone cells with CAG- (n = 5/condition) or GFAP-Cre (n = 10/condition) plasmids along with empty vector or Mash1 plasmids in vivo. **(D)** Experimental paradigm. **(E, F, G)** Quantifications of the number of YFP⁺ cells (E), YFP⁺Mash1⁺ cells (F) and YFP⁺DCX⁺ cells (G). **(H)** Micrograph of a representative YFP⁺DCX⁺ cluster, observed only with Mash1 overexpression. **(I, J)** Quantification of YFP⁺Mash1⁺ cells (I) and YFP⁺DCX⁺ cells (J), combined from both groups of electroporations. Note that Mash1 overexpression increases the expression of the neuroblast marker DCX in electroporated cells. Two-way ANOVA, Tukey's multiple comparison post hoc test. **(B, H)** Scale bar in (B) represents 50 and 25 µm in (H). *P ≤ 0.05 **P ≤ 0.01, ***P ≤ 0.001.

EV (4/2660) to 1.27% of recombined cells with Mash1 (32/2501), an approximately eightfold increase (Fig 8I and J).

Thus, when Mash1 is overexpressed in quiescent VZ cells, a subpopulation of VZ cells up-regulates Mash1 protein and exhibits a neurogenic increase. Given that plasmid-based expression vectors are transient, these data support the testing of more prolonged expression vectors to enhance the neurogenesis originating from the VZ compartment.

# Discussion

Understanding how the NSC pool is maintained under homeostatic and/or pathological conditions has important implications for endogenous brain repair strategies. Cell-sorting strategies have consistently revealed that the V-SVZ niche contains both cycling aNSCs and more dormant precursors such as qNSCs (Codega et al, 2014), pre-GEPCOT cells (Mich et al, 2014), LeX bright cells (Morizur et al, 2018), or "primitive" NSCs (Reeve et al, 2017). Single-cell RNAseq technologies have helped to define and order stages of the neurogenic lineage based on their transcriptional profiles, adding unprecedented depth to our knowledge of the genetic changes involved in NSC lineage progression (Llorens-Bobadilla et al, 2015; Dulken et al, 2017; Mizrak et al, 2019). However, both the extent and the conditions under which dormant neural precursors contribute to the pool of cycling aNSCs in vivo have remained unclear. Here, we focused on ventricle-contacting GFAP⁺ cells of the VZ compartment, using two separate strategies to probe the contributions of these cells to V-SVZ neurogenesis, the aNSC pool

and niche regeneration. Unexpectedly, our data support that at least a subset of ventricle-contacting GFAP$^+$ cells (including FoxJ1$^+$GFAP$^+$ cells) exhibit in vivo neurogenic activity without contributing to the highly expanding, neurosphere-forming pool of aNSCs. Our data shed light on a previously undetected aspect of V-SVZ neurogenesis, adding an additional component to the current model of neurogenic activity within this niche (Fig 9).

## Neurogenesis originating from the ventricular epithelium

It is well established that the ventricular epithelium of the V-SVZ is a source of cells that are capable of neurogenesis in vitro and in vivo (Coskun et al, 2008; Codega et al, 2014; Chaker et al, 2016; Lim & Alvarez-Buylla, 2016; Obernier et al, 2018). It is also generally agreed that, of its two main cell types, B1 astrocytes rather than ependymal cells are at the origin of this neurogenesis (Obernier et al, 2018; Shah et al, 2018). However, key aspects related to the magnitude and dynamics of neurogenesis originating from the adult VZ compartment have remained unclear. Lineage-tracing and neurosphere-based studies have established that the V-SVZ niche contains a population of quiescent, GFAP$^+$ neural precursors that have the ability to generate neurosphere-forming aNSCs and to regenerate the niche following anti-mitotic treatments (Doetsch et al, 1999b, 2002). The VZ compartment may represent the normal location of these quiescent precursors, as cell sorting approaches have shown that quiescent, CD133$^+$GFAP$^+$ precursors (qNSCs) are able to give rise to aNSCs with a delayed time-course in vitro (Codega et al, 2014). However, there are currently few potential methods for specifically lineage-tracing quiescent GFAP$^+$ VZ cells. For example, GFAP$^{CreER^{T2}}$ mice can be used to identify a quiescent V-SVZ population that is capable of aNSC formation and niche regeneration (Mich et al, 2014; Sachewsky et al, 2019), but this transgenic model recombines cells from both VZ and SVZ compartments. Retroviral labeling of GFAP$^+$ cells can restrict recombination to cells in the VZ and yields labeled olfactory neurons (Doetsch et al, 1999a; Ihrie & Alvarez-Buylla, 2008; Obernier et al, 2018); however, because retroviruses incorporate exclusively into dividing cells, they can be used to probe the biology of actively cycling but not quiescent precursors.

Here, we first used an adult brain electroporation strategy to study GFAP$^+$ VZ cells. Electroporation in 3-mo-old mice labeled a highly quiescent subset of 500–600 GFAP$^+$ cells per VZ, composed of about 10% B1 cells and 90% non-B1 GFAP$^+$ cells. After electroporation, these cohorts of recombined cells exhibited limited but stable formation of olfactory neuroblasts for at least 1.5 yr. Gliogenesis was not specifically investigated because of limitations of the model; because the directly electroporated cells themselves express markers of astrocytes and/or ependymal cells, it is not readily possible to distinguish between the electroporated cells and occasional astrocyte/ependymal progeny they might produce within the V-SVZ niche. Although recombined cells were never detected in the corpus callosum or striatum at any time point, as would have been suggestive of oligodendrocyte or striatal astrocyte differentiation, we cannot exclude the possibility of participation in an aging-related replacement of local niche cells. Indeed, whereas no change in YFP$^+$ cell number within the V-SVZ niche was observed between 3 and 8 mo of age, the number of YFP$^+$ cells in the V-SVZ

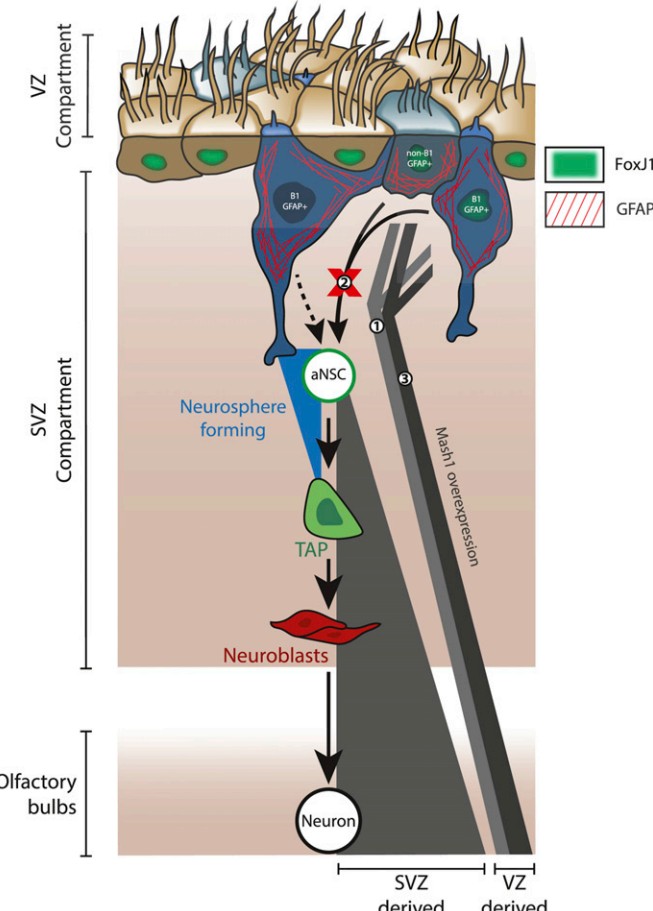

**Figure 9. Summary and proposed relationships between GFAP$^+$ ventricular zone cells and neurogenesis in the adult ventricular–subventricular zone (V-SVZ) niche.**

**(A)** Diagram summarizing main findings. Fate-mapping using hGFAP-driven plasmid electroporation and/or FoxJ1-driven transgenic approaches labeled a population of neurogenic ventricular zone cells that contributes small numbers of neurons to the olfactory bulbs during adulthood (1), yet they do not show evidence of acting via the neurosphere-forming neural stem cell pool (2). These data do not exclude the possible existence of B1 cells that can produce activated neural stem cells in vivo (dotted arrow) but indicate that such cells would be resistant to electroporation and FoxJ1-negative, and that such a capacity is not generalizable to the entire B1 population. Mash1 overexpression promotes a modest increase in neurogenesis by hGFAP-electroporated cells (3); it remains to be determined whether this originates with the B1 or non-B1 GFAP$^+$ cells that are electroporated.

doubled by 21 mo of age. Importantly, despite their neurogenic activity, recombined cells were not recruited during Ara-C–induced niche regeneration and they did not produce neurosphere-forming aNSCs for a period of at least several months of in vivo lineage-tracing, indicating that the neurons to which they give rise are not generated via the neurosphere-forming aNSC pool.

We verified our electroporation findings using a methodologically independent approach. In the V-SVZ, expression of the cilia-associated transcription factor, FoxJ1, is limited to the VZ compartment, where it is expressed in both ependymal cells and a subset of GFAP$^+$ cells (Carlen et al, 2009; Jacquet et al, 2009; Beckervordersandforth et al, 2010). During nervous system

development, FoxJ1-expressing radial glial precursors differentiate into FoxJ1$^+$ ependymal cells and FoxJ1$^+$GFAP$^+$ cells perinatally (Jacquet et al, 2009). The FoxJ1$^+$ population in the early postnatal brain then gives rise to a small subset of OB neurons and harbors neurosphere-forming potential in vitro (Jacquet et al, 2009, 2011). Whether FoxJ1$^+$ population in the adult brain retained this capacity was unclear. We observed that most cells electroporated with hGFAP-driven plasmids were FoxJ1$^+$ on whole-mounts. We, therefore, performed tamoxifen-induced lineage tracing in 3-mo-old FoxJ1$^{CreERT2}$-EYFP mice to test the neurogenic contribution of the adult FoxJ1-expressing population. Immunostaining confirmed that 3–4% of all tamoxifen-induced YFP$^+$ cells were indeed FoxJ1$^+$GFAP$^+$ cells, an estimated 200–300/ventricle at 1-wk post-tamoxifen treatment. At 4 and 16 wk post-tamoxifen, small numbers of YFP$^+$DCX$^+$ neuroblasts were detectable in the V-SVZ, and increasing numbers were present within the OBs. Notably, none of the neurospheres generated from brains at 4 wk post-tamoxifen treatment were recombined for YFP, and a maximum of a *single* YFP$^+$ recombined neurosphere was generated from the brains of mice cultured at 16 wk post-tamoxifen (i.e., at 7 mo of age). Because tamoxifen-induced recombination was extensive (about 50% of all the endogenous FoxJ1-expressing V-SVZ cells) and included 200–300 FoxJ1$^+$GFAP$^+$ cells, contribution of the FoxJ1$^+$ VZ population to the neurosphere-forming aNSC pool is rare during early/mid-adulthood. We did not test the response of the FoxJ1$^+$ VZ population during Ara-C–induced niche regeneration, as we had for electroporated cells; however, a recent study using this lineage-tracing model showed no significant neurogenic response in brain injury/stroke models (Muthusamy et al, 2018). Thus, the FoxJ1-driven recombination model supports our electroporation data, confirming the presence of quiescent VZ precursors that produce neuroblasts without expanding through a neurosphere-forming aNSC intermediate.

The precise identity of the neurogenic VZ cells studied here remains to be determined. The predominant cell types in the VZ are FoxJ1$^+$ ependymal cells and GFAP$^+$ B1 cells. However, FoxJ1$^+$GFAP$^+$ cells also exist (Jacquet et al, 2009; Beckervordersandforth et al, 2010; Codega et al, 2014). In line with this, FoxJ1 protein expression was observed in i) a subset of GFAP protein-expressing cells on coronal sections (Fig 5A), ii) a subset of GFP$^+$ cells in GFAP-GFP mice (Fig 5B), and iii) in an estimated 70% of GFP$^+$ cells in mice electroporated with hGFAP-GFP plasmids. Because mature ependymal cells lack neurogenic activity (Shah et al, 2018), neuroblasts observed in the hGFAP-electroporation and FoxJ1-transgenic models are presumed to be at least partially derived from the FoxJ1$^+$GFAP$^+$ cells. Such FoxJ1$^+$GFAP$^+$ cells may represent a subset of B1 astrocytes and/or non-B1 cells such as astrocyte/ependymal transitional cells (Luo et al, 2006, 2008), E2 ependymal cells (Mirzadeh et al, 2017), or niche astrocytes. It remains to be established whether the sustained production of DCX$^+$ neuroblasts from an initially labeled cohort of VZ cells derives from B1 and/or non-B1 GFAP$^+$ cells, as well as whether it occurs via repeated neurogenic divisions or increasing recruitment from within the pool of initially labeled cells. It will also be interesting to determine the phenotypic identity of the small numbers of OB neurons generated by the electroporated or FoxJ1-traced VZ cells. Prior lineage-tracing of FoxJ1-derived olfactory neurons in the early postnatal brain indicated a contribution to the Tbr1+ glutamatergic population of periglomerular

interneurons, but the identity of the granule neurons that were produced remained undetermined (Jacquet et al, 2011).

## Flux from the adult ventricular epithelium into the neurosphere-forming NSC pool is limited

The studies presented here sought to test the hypothesis that the neurosphere-forming aNSC pool is actively maintained by dormant, ventricle-contacting precursors located within the VZ compartment (Chaker et al, 2016; Lim & Alvarez-Buylla, 2016). Although we did not detect the hypothesized flux from the adult VZ to the aNSC pool, we did demonstrate the presence of cells that exhibit low levels of in vivo neurogenic activity throughout adulthood. Moreover, overexpression of Mash1/Ascl1 increased the neurogenic output of these cells without promoting neurosphere formation. This suggests a model in which there are two pools of neurogenically competent cells within the V-SVZ: neurosphere-forming NSCs in the SVZ and non–neurosphere-forming precursors in the VZ (see Fig 9).

The existence of GFAP$^+$ precursors in the VZ compartment that produce neurons via a minimally amplifying, non-aNSC route in vivo is reminiscent of previous in vitro observations. Using distinct flow cytometry strategies, multiple groups have isolated subpopulations of cells from the V-SVZ that have neurogenic competence but differ in their neurosphere-forming ability; for example, aNSCs versus qNSCs (Llorens-Bobadilla et al, 2015; Dulken et al, 2017) and GEPCOT versus pre-GEPCOT cells (Mich et al, 2014). Although most commonly interpreted as distinct stages within a single neurogenic lineage, it is equally possible that neurosphere-forming and non–neurosphere-forming precursors represent developmentally related precursor lineages that remain distinct during adulthood. The latter interpretation would be consistent with the minimal transcriptomic overlap observed between aNSCs and qNSCs (Llorens-Bobadilla et al, 2015; Dulken et al, 2017; Zywitza et al, 2018).

Interestingly, such a model bears a striking resemblance to embryonic brain development, where two distinct modes of neurogenesis are exhibited by radial glial precursors. "Direct" neurogenesis occurs at the onset of the neurogenic period, when radial glial cells initially produce post-mitotic neuronal daughter cells via asymmetric cell divisions. Direct production of neurons is then rapidly replaced by an "indirect" or amplifying mode of neurogenesis, in which radial glial cells generate proliferative intermediate progenitors that amplify in number before undergoing neurogenesis. For example, primate cortical neurogenesis is initiated by ventricle-contacting radial glial cells (analogous to B1 cells), and then later sustained by a separate subpopulation of "outer radial glia cells" that have withdrawn their apical process and persist in the outer SVZ (Hansen et al, 2010; Nowakowski et al, 2016). Interestingly, B1 cells have recently been observed to give rise to non–ventricle-contacting B2 astrocytes (Obernier et al, 2018).

The demonstration of ventricle-contacting, neuron-producing FoxJ1$^+$GFAP$^+$ precursors that are distinct from the aNSC lineage does not necessarily exclude the existence of other quiescent precursors in the VZ being able to contribute to the aNSC pool. However, such a capacity (if it exists) is clearly not generalizable to all GFAP$^+$ cells in the VZ, and in fact may be limited to a minority of the B1 population. Any such precursors would be predicted to be FoxJ1-negative and inaccessible to electroporated plasmids.

## Summary

Overall, data from the present study reveal that the adult VZ contains quiescent, GFAP+ cells that have neurogenic potential in vivo but that make little (if any) ongoing contribution to the aNSC pool, either under basal or regenerating conditions. This suggests that the adult V-SVZ niche, like the developing brain, has two separate neurogenic pathways. The importance of neurogenic VZ cells as niche components and whether they can be expanded or genetically modified in vivo to serve as an exploitable neurogenic reservoir are important topics that remain to be further explored.

# Materials and Methods

### Contact for reagent and resource sharing

Further information and requests for resources and reagents should be directed to and will be fulfilled by the Lead Contact, Karl Fernandes (karl.jl.fernandes@umontreal.ca).

### Experimental model and subject details

Animal work was conducted in accordance with the guidelines of the Canadian Council of Animal Care and approved by the animal care committees of the University of Montreal and the Research Center of the University of Montreal Hospital (CRCHUM). For these experiments, we used male Rosa26-stop-EYFP (B6.19X1-Gt(ROSA)26Sor$^{tm1(EYFP)Cos}$/J; stock number: 006148), Rosa26-stop-Tom (Gt(ROSA)26Sortm14(CAG-tdTomato)Hze), FoxJ1-Cre$^{ERT2}$GFP (Foxj1$^{tm1.1(Cre/ERT2/GFP)Htg}$/J; stock number 027012), hGFAP$^{CreERT2}$ (B6.Cg-Tg(GFAP-Cre/ERT2)505Fmv/J), and hGFAP::GFP (FVB/N-Tg(GFAPGFP)14Mes/J; stock number: 003257). FoxJ1-Cre$^{ERT2}$GFP mice were crossed with Rosa26-stop-EYFP mice, hGFAP$^{CreERT2}$ mice were crossed with Rosa26-stop-Tomato mice, and tamoxifen induction was performed by two gavages at 750 mg/kg diluted in 9:1 corn oil and ethanol. Mice were socially housed (up to five mice/cage) before surgery with a 12-h light–dark cycle with free access to water and food. Mice were individually housed post-surgery.

### Method details

#### Surgical procedures

Mice received acetaminophen drinking solution (1.34 mg/ml, Tylenol) from 1 d before surgery until 3 d after surgery. Surgeries were performed under isoflurane general anesthesia (Baxter) and bupivacaine local anesthesia (1 mg/kg; Hospira).

**Electroporation** Adult brain electroporation was conducted as previously described (Barnabe-Heider et al, 2008; Hamilton et al, 2015; Joppe et al, 2015). Plasmids (Table 1) were amplified by using an endotoxin-free 40-min Fast Plasmid Maxiprep Kit (Biotool), and then purified and concentrated by ethanol precipitation. Intra-cerebroventricular plasmid injections were performed using a 10 μl Hamilton syringe into the left ventricle at coordinates: 0 mm ante-roposterior (AP), +0.9 mm mediolateral (ML), –1.5-mm dorsoventral (DV) to Bregma. Animals received an ICV injection of 10 μg of total DNA in 2 μl, delivered over 2 min, followed by five pulses at 50-ms intervals at 200 V applied with 7-mm platinum Tweezertrodes (Harvard Apparatus) and an electroporator (ECM 830; Harvard Apparatus). If electroporation was combined with osmotic pump infusion, pumps were implanted contralaterally. Titration experiments were performed to determine the optimal plasmid concentration for electroporations (Fig S3A–C).

**Osmotic pump infusions** ICV infusions were performed using, 7-d osmotic pumps (Alzet, model 1007D; Durect) attached to brain infusion cannula (Alzet, Brain infusion kit 3; Durect). Cannulae were stereotaxically implanted in the right ventricle at coordinates: 0 mm AP and –0.9 mm ML to the bregma. For antimitotic experiments, 2% of Ara-C (Sigma-Aldrich) or vehicle was infused for 7 d, and then animals were either euthanized for immediate analysis or pumps were removed and mice euthanized 21 d later.

#### Tissue analysis

Mice were euthanized by intraperitoneal injection of ketamine/xylazine (347/44 mg/kg; Bimeda-MTC/Boehringer Ingelheim Canada Ltd). For immunostaining, mice were intracardially perfused with PBS (Wisent) followed by freshly prepared 4% paraformaldehyde (Acros). Brains were removed, post-fixed overnight, and then cut into 40-μm sections using a Leica VT1000S vibrating microtome. Tissue sections were stored in antifreeze at –20°C (Bouab et al, 2011). For neurosphere assays, brains were dissected from freshly euthanized mice.

**Immunostaining** Antibodies are listed in Table 2. Immunostaining was performed as described previously (Bouab et al, 2011; Gregoire et al, 2014). Citrate-EDTA antigen retrieval was used for immunostaining with DCX and Ki67 antibodies. BrdU staining of the OBs was performed using HCl denaturation and visualization using DAB (3-3'-diaminobenzide) (Gregoire et al, 2014). Whole-mount stainings were performed as described by Mirzadeh et al (2008).

**Table 1. Plasmid list.**

| Name | Regulatory elements | Gene | Company | Gift of: |
|------|--------------------|------|---------|----------|
| CAG-Cre | CAG | *Cre* | Addgene (# 13775) | Connie Cepko |
| hGFAP-Cre | hGFAP | *Cre* | Addgene (# 40591) | Albee Messing |
| hGFAPmyrGFP | hGFAP | *myrGFP* | Addgene (# 22672) | Robert Benezra |
| hGFAPmyrTomato | hGFAP | *myrTomato* | Addgene (# 22671) | Robert Benezra |
| Mash1 | CMV | *ASCL1* | OriGene (RC201123) | N/A |

**Table 2. Antibody list.**

| Name | Specie | Company | Dilution |
| --- | --- | --- | --- |
| BrdU | Rat | AbDSerotec | 1:800 |
| DCX | Guinea Pig | Chemicon | 1:3,000 |
| DCX | Goat | Santa Cruz Biotech | 1:250 |
| FoxJ1 | Mouse | Sigma-Aldrich | 1:250 |
| GFP | Chicken | Aves Lab | 1:2,000 |
| GFAP | Chicken | Novus Biologicals | 1:1,000 |
| GFAP | Rabbit | Dako Diagnostic | 1:500 |
| GFAP | Mouse | Chemicon | 1:1,000 |
| Ki67 | Mouse | BD Biosciences | 1:100 |
| Mash1 | Mouse | BD Biosciences | 1:50 |
| Olig2 | Rabbit | Chemicon | 1: 250 |
| S100B | Mouse/Rabbit | Sigma-Aldrich | 1:1,000 |
| Sox2 | Rabbit | Chemicon | 1:1,000 |
| $\beta$III tubulin | Mouse | Covance | 1:200 |
| Secondary Ab (Alexa) | Goat, Donkey | Invitrogen | 1:1,000 |

EdU (5-ethynyl-2′-deoxyuridine) staining was performed as described by Salic and Mitchison (2008). Briefly, the sections were incubated in EdU reaction solution (100 mM Tris-buffered saline, 2 mM $CuSO_4$, 4 $\mu$M sulfo-cyanine 3 azide, and 100 mM sodium ascorbate) for 5 min, then washed with two quick washes before to be incubated 5 min in copper blocking reaction (10 mM THPTA in PBS). When EdU staining was coupled with another antibody staining, EdU was performed first.

**Neurosphere assays** Neurosphere cultures were generated from adult mouse striatum using 20 ng/ml EGF (Sigma-Aldrich) and a protocol based on Reynolds and Weiss (1992) as detailed previously (Bouab et al, 2011; Hamilton et al, 2015; Joppe et al, 2015). Cells were fed each week with EGF and B27 (2%; Invitrogen). Factors were added alone or in combination at the following concentrations: EGF 20 ng/ml, FGF-2 10 ng/ml, and LIF 10 ng/ml.

### Quantification and statistical analyses

Immunostained tissue sections were examined using a motorized Olympus IX81 fluorescence microscope, an Olympus BX43F light microscope, a Zeiss Axio Observer.Z1 inverted microscope coupled with a Yokogawa Spinning Disc scanning Unit CSU-X1 (Yokogawa Electric Corporation) or a Leica TCS-SP5 inverted microscope (Leica Microsystems). All quantifications were performed by a blinded observer using coded slides and 40×, 60×, or 100× objectives. For quantification of total Ki67 or DCX cells, 4–6 V-SVZ sections/animal were analyzed. For quantifications of V-SVZ whole-mount preparations, 6–8 fields/animal were analyzed. For quantification of electroporated cells and their progeny, 6–12 sections/animal were used for the V-SVZ and 5–15 sections/animal for the OB. In a control series of animals, we quantified 100% of sections from the SVZ and OB and found no difference versus the above tissue sampling approach, indicating we were not missing small clusters of labeled cells (Fig S4). Counts in the V-SVZ were limited to the DAPI-defined SVZ. Counts in the OB were performed by scanning the entire OB sections for positive cells at 32× objective magnification. All positive cells in the V-SVZ or OB were confirmed for the presence of a DAPI-stained nucleus. For quantification of the dorsoventral distribution of YFP⁺ cells, the counted cells were recorded on a diagram of the ventricle with respect to their position along the dorsoventral axis. The ventricle was divided into four quadrants (dorsal, dorsomedial, ventromedial, and ventral), and the percentage of cells in each quadrant was calculated. Occasional animals presented relatively few electroporated cells and were, thus, considered unsuccessful electroporations; these were excluded from the study. Criteria for exclusion were having less than 20% of the mean number of electroporated cells/section for that experimental group. Of the 280 mice used in this study, a total of eight mice were excluded (two deaths during surgery and six unsuccessful electroporations).

All statistical analyses were achieved using GraphPad Prism, version 6.01 (GraphPad Software, Inc.). Statistical analyses were performed using a two-tailed unpaired $t$ test or one-way or two-way ANOVA with Tukey's post-test, as indicated in figure legends. Error bars represent mean ± SEM. Significance level was set at $P \leq 0.05$.

## Supplementary Information

## Acknowledgements

Spinning-disc and laser-scanning confocal microscopy was performed with the help of the cell imaging core facility of the CRCHUM. This work was supported by studentships from the Université de Montréal Faculty of Graduate Studies (SE Joppé) and the Alzheimer Society of Canada (LK Hamilton) and by grants from the Canadian Institutes of Health Research, the Natural Sciences and Engineering Research Council, and the Canada Research Chairs program (KJL Fernandes).

### Author Contributions

SE Joppé: conceptualization, investigation, visualization, methodology, and writing—original draft, review, and editing.
LM Cochard: conceptualization, investigation, visualization, methodology, and writing—original draft, review, and editing.
L-C Levros: investigation and methodology.
LK Hamilton: investigation.
P Ameslon: investigation.
A Aumont: investigation.
F Barnabé-Heider: conceptualization.
KJL Fernandes: conceptualization, supervision, funding acquisition, visualization, and writing—original draft, review, and editing.

### Conflict of Interest Statement

The authors declare that they have no conflict of interest.

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
