## [Reviewer comments · Life Science Alliance]

Life Science Alliance

Genetic targeting of neurogenic precursors in the adult forebrain ventricular epithelium

Sandra Joppé, Loïc Cochard, Louis-Charles Levros, Laura Hamilton, Pierre Ameslon, Anne Aumont, Fanie Barnabé-Heider, and Karl Fernandes

DOI: <https://doi.org/10.26508/lsa.202000743>

Corresponding author(s): Karl Fernandes, University of Montreal

Review Timeline:

Submission Date:	2020-04-16
Editorial Decision:	2020-04-17
Revision Received:	2020-04-19
Editorial Decision:	2020-05-05
Appeal Received:	2020-05-06
Editorial Decision:	2020-05-11
Revision Received:	2020-05-16
Accepted:	2020-05-18

Scientific Editor: Andrea Leibfried

Transaction Report:

Please note that the manuscript was previously reviewed at another journal and the reports were taken into account in the decision-making process at Life Science Alliance.

Referee #1 Review

Report for Author:

In this manuscript, two approaches are used to label a population of non-cycling adult GFAP(+) cells in direct contact to the ventricle. Authors perform fate mapping analysis as well as in vitro neurosphere culture, to study the NSCs properties of these cells. These experiments, at short and long survival times, as well as following Ara-C treatment, fail to identify these cells as dormant NSCs.

In general, the paper is well written and results are well presented and convincing. Furthermore, the identification of markers of subtypes of NSCs (dormant, vs. quiescent or activated), is particularly timely. The manuscript however suffers from a lack of quantitative data. Indeed, both the rate of recombination in FoxJ1 animals, and the number of electroporated cells are not quantified, which makes it difficult to conclude on the respective contribution of both labelled cells populations to adult neurogenesis.

Please find additional comments below:

Authors first use FoxJ1CreERT mice to fate map a population of FoxJ1(+)/GFAP(+) cells that reside in close proximity of the ventricle. They show that these cells contribute to the production of a very small number of neurons into adulthood, and only marginally contribute to neurosphere formation. This is not a particularly novel finding. Indeed, a previous study (Jacquet et al., 2011) performed similar fate mapping experiments (although at younger ages) to demonstrate that FoxJ1(+) cells within the SVZ/RMS/OB only transiently contribute to OB neurogenesis during the perinatal period, and stop doing so at later postnatal stages. Here authors extend these findings to adulthood and perform long survival time experiments, which rules out the possibility that those cells gradually get reactivated through adult life. Several comments can be made on this first part of the manuscript:

- Jacquet et al., 2011 should be referenced and discussed.
- Are FoxJ1(+)/GFAP(+) cells observed throughout the SVZ or enriched within the dorso-lateral corner, as suggested by Jacquet et al. 2009 and 2011
- The rate of recombination of FoxJ1CreERT mice should be specifically assessed within FoxJ1(+)/GFAP(+), as it might be negligible compared to those of ependymal cells that express higher FoxJ1 expression levels (see Fig 2B)
- As previously reported FoxJ1-derived neurons are sparse and of unknown identity. A characterisation of FoxJ1+ neurons in the OB (e.g. Fig 2G), i.e. morphology, marker expression would have been of interest.
- Was gliogenesis investigated in those animals?

Authors then turn to electroporation of the adult SVZ. Electroporation of hGFAP driven plasmids allow them to target FoxJ1(+) but also 30% of FoxJ1(-)/GFAP(+) cells, to reach essentially similar conclusions. An interesting experiment is the demonstration of that electroporated cells do not contribute to niche regeneration following Ara-C treatment. Several comments can be made on this second part of the manuscript:

- The overall number of electroporated cells (i.e. mostly ependymal cells) is quantified. Similar quantifications should be performed for GFAP(+) cells, as it is currently unclear if their number is marginal or significant, and if it varies between animals. Experiments presented in Suppl Figure 1 rather suggest that a very limited number of GFAP(+) cells is labelled by using this approach.
- Variability of the electroporation efficiency is indeed suggested by Fig5I (and to a lesser extent by Fig5F/G). At 4 and 12 WPE, 2 animals show very high number of YFP+ cells within the OB, when compared to other animals. Was a specific SVZ regions electroporated within these animals? For all these experiments, the electroporation efficiency should be quantified in order to normalize those results.

Referee #2 Review

Report for Author:

In this manuscript, authors selectively targeted potential NSCs in the ventricular zone (VZ) of the mouse lateral ventricles, by two independent methodologies. First, they utilized the Foxj1-CreERT2 knock-in strain, and they found the small number of Cre-reporter expressing cells displayed the GFAP-GFP expression and produced DCX+ neuroblasts/OB interneurons. Second, they targeted GFAP-positive cells in the VZ by electroporation, and authors observed the limited contribution of these targeted VZ cells to neurogenesis in vivo. Interestingly, these marked cells by the both methods did not produce neurospheres, suggesting their lack of NSC properties or deeply quiescent future in vitro. In the last part of manuscript, they mis-expressed proneural gene *Ascl1/Mash1* in the VZ cells marked by the electroporation method, and showed enhanced

neurogenesis.

From these results, authors claimed that in the adult V-SVZ niche, apart from the NSC pools actively producing neurons, distinct type of GFAP-expressing cells exist and contribute to neurogenesis at the lower ratio. In addition, they provided a model that these ventricle-contacting GFAP+ cells are on the different lineage from the majority SVZ NSCs producing neurons in the normal or injured state.

Since neurogenesis from these GFAP-expressing cells can be enhanced by the artificial expression of the proneural transcription factor, they suggested their possibility as a source of neuronal regeneration.

However, from this reviewer's opinion, authors did not provide enough experimental data to support their claims. For instance, in Codega et al. Cell Rep (2014), they carefully described anatomical and molecular features of the ventricle-contacting GFAP+ cells. They found three sub-populations in the GFAP+ cells; GFAP+CD133+, GFAP+CD133+EGFR+, and GFAP+. They showed two populations (GFAP+CD133+ and GFAP+) are quiescent and rarely contribute to the initial neurosphere formation. However, once activated in vitro, they can behave similarly to active NSCs and efficiently generate secondary or tertiary neurospheres.

Authors should more carefully analyze the targeted ventricle-contacting GFAP+ cells, for example by whole-mount 3D analysis of V-SVZ regions, and validate anatomical locations in the well-characterized pinwheel organizations. They can also address NSC-properties by transplantation models. In the in vitro validation, authors should analyze behaviors of the targeted cells in more long-term analysis by multiple passages of the partially formed neurospheres, because quiescent NSCs take significant time, typically one to two weeks, before activation. In the above previous paper, in addition to the neurosphere method, the monolayer NSC culture method was also applied to more precisely characterize the ventricle-contacting GFAP+ cells. As the expression plasmid having mP2 (Prominin-1: CD133) promoter is available, authors should more carefully compare their results with the previous literatures before reaching the conclusion to claim a novel lineage in the adult neurogenesis.

Referee #3 Review

Report for Author:

Review of Joppe et al, "Genetic targeting of neurogenic precursors in the ventricular epithelium of the adult forebrain"

Joppe et al attempt to better understand the pathway of neurogenesis in the adult mouse forebrain VZ/SVZ, by performing cell lineage-tracing and gene overexpression experiments. First, using FoxJ1-CreERT2 transgenic mice they traced descendants of FoxJ1-expressing cells. Second they used electroporated hGFAP-Cre plasmid to trace descendants of hGFAP transgene-expressing cells. Finally they overexpressed Ascl1/Mash1 in either all cells (CAG-Cre plasmid) or VZ/SVZ GFAP-expressing cells (hGFAP-Cre plasmid) and assessed the consequences on neuroblast formation.

The general idea of developing ways to better understand neural stem cell potential and perturbation techniques are useful. But the experiments and presentation require much editing to become widely believable and publishable knowledge.

Critical points.

1) Mislabeling and questionable interpretation of NSCs. Figure 1 and Figure 8 and the text both call the NSCs the neurosphere-forming cells. Multiple papers (Morshead 1994, Doetsch 1999, Pastrana 2009, Codega 2014, Mich 2014, Mizrak 2019) have now established that the NSCs have at least two subpopulations, qNSCs which are B1 cells that don't divide and don't make neurospheres, and aNSCs that do divide and do form neurospheres. The text and figures should represent this fact. The general understanding of the field is that somehow the qNSCs give rise to the aNSCs from lineage tracing experiments (Mich 2014, others), and so Figure 8 of this paper directly contradicts these ideas (see point 2 next). The fact that neurogenesis and aNSCs are both lost during anti-mitotic treatment (as seen by multiple authors, including Fig. 6D of the current manuscript) strongly suggests that neurogenesis is linked to dividing aNSCs.

2) Low recombination rate by FoxJ1-CreERT2 mice. Muthusamy 2014, Figure 5C-D shows much more efficient production of OB neuroblasts and neurons from FoxJ-CreERT2 recombination (using the same Cre-ERT2 line) than the current manuscript. This is probably due to Joppe having used Rosa26-stop-EYFP reporter gene, instead of the more sensitive (both more sensitive to Cre, as well as higher transgene expression in recombined cells) Ai14 Rosa26-stop-tdTomato reporter line. As a result the low number of labeled OB cells and neurons in Fig 2F-G is probably less than it should be, and the numbers of SVZ labeled neurospheres are less than they should be.

It is essential for one of the main points being argued in this manuscript, that "The FoxJ1+ VZ cells do not contribute significantly to the neurosphere-forming NSC pool", that the authors characterize how efficient or inefficient they are recombining these cells. This could be done either via careful microscopy, or by using flow cytometry strategies (Codega 2014, Mich 2014). Most likely only a small minority of qNSCs are being labeled in these mice. Whether there is recombination in spinal cord neurosphere-initiating cells, or in VZ ependymal cells, is totally irrelevant to the central point of these experiments. It would be interesting if FoxJ1-CreERT2 only labels a subset of qNSCs and that those are somehow special, but unfortunately there are no data to address this question. In fact it appears from a recent RNA-seq study (Mizrak 2019, Figure 2B) that FoxJ1 is expressed at a low constant level throughout the SVZ astrocytes/qNSCs, not in a heterogeneous fashion.

3) Interpreting recombination in the plasmid electroporation model. Figure 5B shows that the YFP+ fraction of the Tomato+ cells is ~100%, but that measurement is totally irrelevant. The important number is the YFP+ fraction of GFAP+ cells, which again should be measured by careful microscopy or flow cytometry.

Related to this, the neurosphere assay after plasmid transfection suggests that all the recombined cells do not give rise to neurosphere-initiating cells. In particular the GFAP-cre experiment in Fig. 5L directly contradicts Mich 2014 Fig 3A that demonstrated GFAP-CreERT2-recombined qNSCs give rise to neurospheres over time. This is a dramatic discrepancy that must be explained for the main conclusions of the paper to hold. If the electroporation technique selects for a particularly transfectable subset of GFAP-expressing cells, then the authors need to provide additional data that describe these cells. Are they the aNSCs?

Related to this, the Mash1 experiment would be more interesting if they could get it into the actual qNSCs, and compare the effects of Mash1 overexpression in that cell type, to the effects of overexpression in the undefined transfected cells (aNSCs?). Maybe pro-neural factors can only have effects in dividing cells.

Minor point

1) Why is neuroblast formation so high at 21 months of age? Most studies have reported a loss of neuroblast formation in old age, typically consistent with the loss of neurogenesis in OB seen in Fig 5I. Related, the data at the 4 and 6 month timepoints in Fig 5I is shockingly high variance.

April 17, 2020

Re: Life Science Alliance manuscript #LSA-2020-00743-T

Dr. Karl J Fernandes
University of Montreal
Neurosciences

Dear Dr. Fernandes,

Thank you for transferring your manuscript entitled "Genetic targeting of neurogenic precursors in the ventricular epithelium of the adult forebrain" to Life Science Alliance. The manuscript was assessed by expert reviewers at another journal, and the editors transferred those reports to us with your permission.

The reviewers who evaluated your work elsewhere appreciated various aspects of your study, but thought that your conclusions were not sufficiently supported. We would be happy to consider a revised version of your work and based on the reports already at hand. We would expect a point-by-point response to all concerns and accordingly changes to the manuscript text and data representation, with careful discussion to address the concern of conflicting data (rev#3). Furthermore, the missing quantifications (rev#1) need to get provided, the recombination rate needs to get analyzed (rev#1 and #3), and the electroporation efficiency needs to get analyzed and quantified (rev#1 and #3).

Thank you for this interesting contribution to Life Science Alliance. We are looking forward to receiving your revised manuscript.

Sincerely,

Andrea Leibfried, PhD
Executive Editor
Life Science Alliance
Meyrhofstr. 1
69117 Heidelberg, Germany
t +49 6221 8891 502
e a.leibfried@life-science-alliance.org

B. MANUSCRIPT ORGANIZATION AND FORMATTING:

Dear Dr Leibfried,

I would like to thank the previous reviewers for their constructive and insightful criticisms. We are submitting this revised manuscript to Life Science Alliance, with the goal of using the past reviews to accelerate potential publication. In response to the reviewer comments, we have performed **extensive re-analyses of our pre-existing samples**, produced entirely **new cohorts of animals** for in vitro and in vivo studies, and obtained **new transgenic lines** for experimentation and direct comparison with previously published data (GFAP-CreERT2 mice; Ai14 Tomato reporter mice). Revisions made in response to the reviewers' comments have strengthened the quality and interpretations of our study, which we firmly believe makes a significant and original contribution to the area.

1. (Rev #1) Reference and discuss Jacquet et al

- Response: We agree these are relevant studies to cite.

- Changes: Text added to the Discussion, pages 17:

“During nervous system development, FoxJ1-expressing radial glial precursors differentiate into FoxJ1⁺ ependymal cells and FoxJ1⁺GFAP⁺ cells perinatally (Jacquet et al., 2009). The FoxJ1⁺ population in the early postnatal brain then gives rise to a small subset of olfactory bulb neurons and harbors neurosphere-forming potential in vitro (Jacquet et al., 2011; Jacquet et al., 2009). Whether FoxJ1⁺ population in the adult brain retained this capacity was unclear.”

2. (Rev #1) Are FoxJ1+GFAP+ cells observed through the SVZ or enriched within the dorso-lateral corner... The rate of recombination of FoxJ1CreERT mice should be specifically assessed within the FoxJ1(+)/GFAP(+).

(Rev #3) Low recombination rate by FoxJ1-CreERT2 mice. Muthusamy 2014, Figure 5C-D shows much more efficient production of OB neuroblasts and neurons from FoxJ-CreERT2 recombination (using the same Cre-ERT2 line) than the current manuscript. This is probably due to Joppe having used Rosa26-stop-EYFP reporter gene, instead of the more sensitive...Ai14 Rosa26-stop-tdTomato reporter line. As a result the low number of labeled OB cells and neurons in Fig 2F-G is probably less than it should be, and the numbers of SCZ labeled neurospheres are less than they should be. It is essential for one of the main points being argued in this manuscript that “The FoxJ1+ VZ cells do not contribute significantly to the neurosphere-forming pool”, that the authors characterize how efficient or inefficient they are recombining these cells.

- Response: Reviewer 1 asks for clarification about the extent and location of recombination achieved in the FoxJ1 lineage tracing model, while Reviewer 3 questions why our recombination is much lower than observed in the Muthasamy et al. article and suggests that the lack of recombined neurospheres may be due to the weaker EYFP reporter intensity.
- Changes to address Reviewer 1 and 3: we have now performed studies to evaluate the number and location of FoxJ1+GFAP+ cells that are recombined in our model. A cohort of FoxJ1-CreERT2-EYFP mice were tamoxifen-treated and analyzed 1 week later. First, immunostaining for FoxJ1 protein versus YFP expression demonstrated that the **overall YFP recombination rate** was $52.1 \pm 2.56\%$ of all FoxJ1 protein-expressing cells. Second, we acknowledge the possibility that recombination efficiency might be lower within the GFAP+ subpopulation of YFP+ cells, and so we quantified the absolute number and the location of YFP+GFAP+ cells along the dorsal to ventral axis of the V-SVZ. We found there were 6.9 ± 0.6 YFP+GFAP+ cells/ventricle/section (3 ventricles/animal, N=4 animals), and by extrapolation, an estimated 200-300 YFP+GFAP+ cells per lateral ventricle. These YFP+GFAP+ cells represented $3.95 \pm 0.16\%$ of the YFP+ population (850-1152 YFP+ cells analyzed/animal, N=4 animals), and they were equally spread across dorsal-to-ventral quadrants of the lateral ventricles. These new data have been added to the Results section (pages 10-11) and incorporated as new data panels in Figure 5 (D-H).
- To address other points from Reviewer 3: The Muthusamy article indeed uses the same Cre-ERT2 line, but recombination was induced at a **developmental** time point (Tamoxifen at P0) and analyzed at P21. That article therefore illustrates the neurogenic potential of FoxJ1+ cells in the developing, not adult, brain. With regard to the EYFP reporter intensity, it is indeed weaker than the Ai14 Tomato intensity, but

this does not explain the lack of recombined neurospheres or olfactory bulb neurons. First, we were easily able to detect that half of the neurospheres generated from the spinal cord were EYFP+ (Figure 6C). Second, when we obtained Ai14 Tomato reporter mice and used them for electroporation with hGFAP-cre, recombined neurons in the OB after 1 month were much brighter but only marginally increased in number to what we observed with EYFP reporter mice.

3. (Rev #1) A characterisation of FoxJ1+ neurons in the OB ... would have been of interest.

- Response: We agree this is an interesting question. However, due to the relatively low numbers of FoxJ1-derived OB neurons that were produced in adulthood, this is not possible to test using our existing cohorts of animals. In this regard, we note that even in the Jacquet articles referenced above, the authors were only able to phenotype a minority of the OB neurons derived from FoxJ1+ SVZ cells in the postnatal brain, even though there were substantially greater numbers of recombined cells in that younger model. Since the principal focus of the present article is the V-SVZ niche, we consider this interesting question to be outside the scope of the current study.
- Changes: The following text was added to the Discussion (page 18-19):

“It will also be interesting to determine the phenotypic identity of the small numbers of olfactory bulb neurons generated by the electroporated or FoxJ1-traced VZ cells. Prior lineage-tracing of FoxJ1-derived olfactory neurons in the early postnatal brain indicated a contribution to the Tbr1+ glutamatergic population of periglomerular interneurons, but the identity of the granule neurons that were produced remained undetermined (Jacquet et al., 2011).”

4. (Rev #1) Was gliogenesis investigated in those animals?

- Response: We agree this is an interesting question, however gliogenesis was not specifically investigated due to limitations of the model (below).
- Changes: The following text was added to the Discussion (page 16-17):

“Gliogenesis was not specifically investigated due to limitations of the model: since the directly electroporated cells themselves express markers of astrocytes and/or ependymal cells, it is not readily possible to distinguish between the electroporated cells and occasional astrocyte/ependymal progeny they might produce within the V-SVZ niche. Although recombined cells were never detected in the corpus callosum or striatum at any timepoint, as would have been suggestive of oligodendrocyte or striatal astrocyte differentiation, we cannot exclude the possibility of participation in an aging-related

replacement of local niche cells. Indeed, while no change in YFP⁺ cell number within the V-SVZ niche was observed between 3 and 8 months of age, the number of YFP⁺ cells in the V-SVZ doubled by 21 months of age.”

5. (Rev #1) The overall number of electroporated cells (i.e. mostly ependymal cells) is quantified. Similar quantifications should be performed for GFAP(+) cells, as it is currently unclear if their number is marginal or significant...

(Rev #2) Authors should more carefully analyze the targeted ventricle-contacting GFAP+ cells, for example by whole-mount 3D analysis of V-SVZ regions

(Rev #3) Interpreting recombination in the plasmid electroporation model... The important number is the YFP+ fraction of GFAP+ cells.

- Response: GFAP+ cells in the VZ are typically associated with B1 astrocytes, the cell population that includes both qNSCs and aNSCs. However, the VZ also contains subsets of GFAP+ cells that are distinct from B1 astrocytes. Unlike B1 cells (whose cell bodies are below the ependymal layer and extend a process to the ventricular surface), these non-B1 GFAP+ cells have cell bodies intercalated within the ependymal layer itself. Virtually nothing is known about the biology of non-B1 GFAP+ cells at the ependymal surface. However, they may correspond to a transitional cell type between B1 cells and ependymal cells (Luo et al., JNeurosci, 2008), to GFAP+ type E2 ependymal cells (Mirzadeh et al., Nat Commun, 2017) and/or to niche astrocytes integrated in the ependymal layer. These cells are easily observed in images from recent publications (Luo et al., JNeurosci, 2008; Mirzadeh et al., Nat Commun, 2017; Habela et al., JNeurosci 2020, Figure 1C, Figure 2A; Kokovay et al., Cell Stem Cell 2012, Figure 2A).
- In response to the reviewer questions, we used wholemount preparations, immunostaining and confocal imaging to study the hGFAP-expressing electroporated population in greater detail. Analysis of wholemounts at 1-week post-electroporation revealed two distinct morphologies of hGFAP+ electroporated cells: cells having the morphology of B1 cells (subependymal cell bodies with extensions to the ventricular surface) and cells having cell bodies at the ventricular surface. When we examined the wholemounts of naïve (non-electroporated) mice using GFAP immunofluorescence, we likewise observed these two cell morphologies, similar to seen in the publications referenced above. Since 93% of hGFAP+ electroporated cells expressed GFP reporter when electroporated in the VZ of hGFAP::GFP transgenic mice, **we conclude that the vast majority of the 500-600 hGFAP+ cells we electroporate per brain are indeed GFAP+ VZ cells and that they include both B1 (10%) and non-B1 (90%) subsets of GFAP+ cells.**
- Changes: We have modified the introductory Figure 1 and summary Figure 9 to reflect the presence of non-B1 GFAP+ cells in the VZ. Our interpretations have been also been

modified accordingly, to clarify that the inability of the electroporated subset of GFAP⁺ VZ cells to give rise to neurosphere-forming aNSCs does not exclude the potential ability of B1 cells/qNSCs to do so. The analyses above have also been added to Figure 2, and the Figure 2 Results section has been modified to state the following (page 6):

“Immunofluorescence analysis of lateral ventricle wholemount preparations further confirmed that the proportion of electroporated cells that were clearly positive for high levels of GFAP protein was increased 5-6 fold when using plasmids driven from the hGFAP promoter than from a non-cell-specific promoter (Fig. 2E-F). When we electroporated hGFAP-myrTom (myristolated Tomato) plasmids into the VZ of hGFAP::GFP transgenic mice, 93% of the Tom⁺ cells were indeed GFP⁺ (Fig. 2G). Collectively, these data suggest that the vast majority of the 500-600 cells expressing hGFAP-driven plasmids per electroporated ventricle are indeed GFAP-expressing VZ cells.

The population of GFAP⁺ VZ cells includes both B1 astrocytes (whose cell bodies are below the ependymal layer and extend a process to the ventricular surface)(Kokovay et al., 2012; Mirzadeh et al., 2008) and subsets of non-B1 GFAP⁺ cells (whose cell bodies are intercalated within the ependymal layer itself) (Habela et al., 2020; Luo et al., 2008; Mirzadeh et al., 2017). Consistent with this, wholemount analysis of the ventricular walls of mice electroporated with hGFAP-myrTom plasmids revealed two distinct morphologies: 9.3±1.7% of Tom⁺ cells had the small, subependymal cell body with one or more branches characteristic of B1 cells (Fig. 2H, top) and 90.7±1.7% had larger cell bodies located directly at the ventricular surface (Fig. 2H, bottom) (6-7 fields from each of 2 wholemounts). Immunostaining for GFAP protein on wholemounts from naive control mice confirmed that GFAP⁺ cells having these morphologies are likewise present in the non-electroporated VZ (Fig. 2I).

Thus, using adult brain electroporation, we target 500-600 GFAP⁺ VZ cells per electroporated ventricle, with the majority being non-B1 GFAP⁺ cells.”

6. (Rev #1) At 4 and 12 WPE, 2 animals show very high number of YFP⁺ cells within the OB, when compared to other animals. Was a specific SVZ regions electroporated within these animals?

(Rev #3) Minor point... the data at the 4 and 6 month timepoints in Fig 5I is shockingly high variance.

- Response: Reviewers 1 and 3 are referring to the same two animals. First, it should be noted that the 31 animals in this timecourse experiment (Fig. 3F-I) were all electroporated in the same surgery session and with the same plasmid preparations, and with the exception of the two animals in question, the data points at each timepoint are actually fairly tight for such a complex *in vivo* manipulation. To understand whether or not these two animals had a different regional distribution of electroporated cells, we recorded the dorso-ventral location of all recombined cells from 6 sections/animal of all animals within the 4 and 12 WPE timepoints. Importantly, we found no dorso-ventral or anterior-posterior difference in regional distribution of the recombined cells between the two outlier animals and the other animals in their group. Neither did we observe any unusual clustering of cells that might suggest an exceptional electroporation of an amplifying aNSC. Thus, it appears that these two animals simply had a greater number of initially electroporated cells (as might occur if a greater volume of plasmid was injected) rather than a regionally-specific electroporation pattern.
- Changes: Although some authors would exclude these two points as outliers, there is no way to retrospectively determine the number of initially electroporated cells, and so we see no valid way of justifying their exclusion. Instead, we have included them for transparency, and also included the above detailed analyses of these two animals in Supplementary Figure 2 (referred to from page 7 of the Results):

“It was noted that two of the thirty-one animals in this timecourse experiment had highly elevated numbers of recombined cells (1/4 mice at 4 WPE and 1/5 mice at 12 WPE): more detailed analysis of these two animals revealed no clusters or differences in spatial localization of recombined cells, suggesting they simply had more cells initially electroporated (supplementary figure 2).”

7. (Rev #2) They can also address NSC-properties by transplantation models

- Response: Agreed that a transplantation approach might also be informative. However, performing well-controlled FACS and transplantation studies on small numbers of electroporated cells is outside the scope of the present study.

8. (Rev #2) Authors should analyze behaviors of the targeted cells in more long-term analysis by multiple passages of the partially formed neurospheres, because quiescent NSCs take significant time, typically one to two weeks, before activation. In the above previous paper, in addition to the neurosphere method, the monolayer NSC culture method was also applied to more precisely characterize the ventricle-contacting GFAP+ cells.

- Response: Agreed. To do these experiments, we used both EYFP and Tomato reporter mice, to ensure the weaker EYFP signal would not result in false negatives, and obtained similar results for both. Reporter mice were electroporated using the hGFAP-cre promoter as previously, and after waiting one week we generated neurosphere cultures. After two weeks of neurosphere growth, when electroporated cells had again generated large numbers of small colonies and zero neurospheres, the cultures were passaged with regular re-feeding. Secondary neurospheres grew easily from these cultures, as typically observed, but none of these secondary neurospheres were fluorescent, and the fluorescent cells in fact disappeared from these passaged cultures. We also generated dissociated monolayer cultures from the primary cultures of neurospheres and fluorescent colonies, but similar to the secondary neurosphere cultures, electroporated cells in the monolayers did not proliferate or persist. Thus, the hGFAP electroporated cells have minimal proliferative ability and clearly behave distinctly from the FACS-isolated qNSCs reported in Codega et al., 2014.
- Changes: This new experiment has been added to the Results section (page 9) and to Figure 4C.

9. (Rev #3) Mislabeling and questionable interpretation of NSCs.

- Response: Agreed. Moreover, based on the new anatomical data (point 5 above) and cell culture data (point 8 above), we can further specify that our electroporation procedure preferentially labels a population that is distinct from both qNSCs and aNSCs.
- Changes: We have now distinguished between qNSCs and aNSCs throughout the revised manuscript and figures. We also clarify that our proposed model does not exclude the possibility that B1/qNSCs cells give rise to aNSCs, since electroporation preferentially targets GFAP+ cells that are *not* B1 cells (point 5).

10. (Rev #3) The general understanding of the field is that somehow the qNSCs give rise to the aNSCs from lineage tracing experiments (Mich 2014, others), and so Figure 8 of this paper directly contradicts these ideas.... In particular the GFAP-cre experiment in Fig. 5L directly contradicts Mich 2014 Fig 3A that demonstrated GFAP-CreERT2-recombined qNSCs give rise to neurospheres over time. This is a dramatic discrepancy that must be explained for the main conclusions of the paper to hold. If the electroporation technique selects for a particularly transfectable subset of GFAP-expressing cells, then the authors need to provide additional data that describe these cells. Are they the aNSCs? Related to this, the Mash1 experiment would be more interesting if they could get it into the actual qNSCs.

- Response: We absolutely agree with Reviewer 3 (that Mich et al. use lineage tracing to show that the SVZ contains qNSCs that can give rise to neurosphere-forming aNSCs over time). However, what Reviewer 3 describes as a “dramatic discrepancy” is in fact the central reason we performed this study. The GFAP-CreERT2 mice used in Mich et al recombines in ALL GFAP+ cells throughout the SVZ and VZ compartments, while the electroporation and FoxJ1-based approaches used in our study recombine solely in cells within the VZ compartment. There is no discrepancy.
- To emphasize this point, we obtained the GFAP-CreERT2 mice used in Mich et al., crossed them with Ai14 Tomato reporter mice, and then induced recombination through either Tamoxifen treatment (as done by Mich et al.) versus hGFAP-Cre electroporation. Tamoxifen treatment yielded the same results as in Mich et al. (recombination in GFAP+ cells throughout the V-SVZ niche and fluorescent neurospheres). Conversely, electroporation yielded results as reported in our manuscript (recombination restricted to GFAP+ cells in the VZ and an absence of recombined neurospheres).
- Changes: These new experiments using GFAP-creERT2-Tom mice have been added to a revised Figure 1 in order to make clear a key objective of the article: to study the neurogenic properties of GFAP+ cells within the VZ compartment. They are described on page 5 of the Results as follows:

“To test whether adult brain electroporation would restrict recombination to cells in the VZ compartment, we used Tamoxifen-inducible hGFAP^{CreERT2} transgenic mice crossed with Rosa26-stop-Tomato reporter mice (the cross herein referred to as hGFAP^{CreERT2}-Tom mice). Recombination in hGFAP^{CreERT2}-Tom mice was induced by either Tamoxifen treatment (for global recombination in GFAP⁺ cells) or by electroporation of hGFAP-driven Cre-recombinase plasmid (hGFAP-cre; for local recombination in the VZ) (Fig. 1C). After 7 days, Tamoxifen-treated mice had recombined Tom⁺ cells throughout the V-SVZ niche, as expected (Fig. 1D). In contrast, hGFAP-cre - electroporated mice had recombined cells only adjacent to the ventricular surface (Fig 1E). In mice processed for neurosphere cultures, recombined aNSC-associated neurospheres were observed in the Tamoxifen-treated mice as previously reported (Mich et al., 2014), but not in hGFAP-cre electroporated mice (Fig 1F-G).”

May 5, 2020

Re: Life Science Alliance manuscript #LSA-2020-00743-TR

Dr. Karl J.L. Fernandes
University of Montreal
Neurosciences
900 rue St-Denis,
CRCHUM - Tour Viger, Neuroscience Dept, R09-482
Montreal, Quebec H2X 0A9
Canada

Dear Dr. Fernandes,

Thank you for submitting your revised manuscript entitled "Genetic targeting of neurogenic precursors in the adult forebrain ventricular epithelium" to Life Science Alliance.

We contacted the original reviewers that provided their identity to us to enable efficient re-review of your work. Reviewer #1 was not willing to look again at your work, but reviewer #3 has now provided comments on the revised version. We had informed the reviewer about the transfer situation and on our expectations regarding the extent of a revision necessary for publication here. Unfortunately, and as you can see below, while the reviewer recognizes the effort that went into clarifying a lot of the aspects previously criticized, s/he does not think that the revised version addresses the revision requirements in a satisfactory way. The reviewer therefore does not support publication. Given the lack of support on the revised version, we unfortunately cannot offer publication of the manuscript. I realize that this is very disappointing and I am sorry that I couldn't bring better news.

Best wishes,
Andrea

Reviewer #1 (Comments to the Authors (Required)):

In this revised manuscript, authors tried well to clarified their methodologies and findings in the current understanding of the neural stem cells and the mode of neurogenesis at the ventricular-subventricular zone of the mouse lateral ventricle.

However, this reviewer hesitates to accept their claims about the novel GFAP-promoter positive neural stem cell population and its contribution to neurogenesis in the adult brain.

First, they applied HUMAN GFAP promoter plasmids to express Cre recombinase for lineage tracing and fluorescent proteins for validating the electroporated cells in the MOUSE brain. In addition, the activity of GFAP promoter is reportedly regulated by epigenetic modifications, such as DNA methylation on its promoter (REF: <https://www.ncbi.nlm.nih.gov/pubmed/11740937>). The transient introduction of the expressions plasmids with electroporation is not suitable for conducting strict lineage-trace experiments when these regulatory mechanisms on the GFAP promoter are considered.

Secondly, as the other reviewer criticized, FoxJ1 is expressed in a subset of Type-B1 cells. This makes interpretation of the results of FoxJ1-CreERT2 mice quite intricate.

Thirdly, the number of Non-B1 GFAP cells and its contribution to adult neurogenesis seem to be extremely limited. Considering this limited existence of this population, the schematic representation in Figure 9 is not a suitable image to illustrate the ventricular-subventricular zone neurogenesis in the adult brain.

The authors of manuscript #LSA-2020-00743-TR have requested an appeal. Their comments are below.

Dear Andrea,

Thank you for your email. However, it is indeed very unfortunate that the editorial decision is based only on Reviewer #3. As I had (gently) pointed out in our response letter, Reviewer #3 had already made several important errors in the first round of reviews, in their interpretations of the previous literature (article by Mich et al. and article by Muthusamy et al, discussed in points 2 and 10 of our response letter). Here in their new comments, Reviewer #3 has again made major errors:

1. Reviewer #3 criticizes our use of the HUMAN GFAP promoter. However, this is the identical promoter used in previously published articles by many groups using hGFAP-GFP transgenic mice to isolate NSCs (beginning with Codega et al., Neuron 2014). The hGFAP promoter is the promoter used for this purpose in the field.
2. Reviewer #3 further criticizes the use of transient expression plasmids for lineage tracing. We do not use transient expression plasmids for lineage tracing, we use them to induce permanent Cre-mediated recombination, which is then maintained in all progeny regardless of any subsequent regulation at the GFAP promoter.

The presence of FoxJ1 in a subpopulation of B1 cells (which we directly show and quantify), and the fact that both B1 and non-B1 GFAP+ cells are labelled, are factors that can impact only the interpretation of the data (which we have done very carefully).

With respect, we are unfairly and severely disadvantaged by having the fate of this manuscript decided only by Reviewer #3, especially after performing 5 months of revisions. Do we have any recourse to appeal this decision? Is it possible to have a timely assessment by an unbiased reviewer? Or by Reviewer #2?

MS: LSA-2020-00743-TR

Dr. Karl J.L. Fernandes
University of Montreal
Neurosciences
900 rue St-Denis,
CRCHUM - Tour Viger, Neuroscience Dept, R09-482
Montreal, Quebec H2X 0A9
Canada

Dear Dr. Fernandes,

Thank you for your recent correspondence regarding our decision on your revised manuscript "Genetic targeting of neurogenic precursors in the adult forebrain ventricular epithelium". I read your arguments and decided that it was indeed warranted to obtain an independent opinion on the revised version in this case. We thus reached out to the original reviewer again, who now kindly agreed to re-review your work. I copy this reviewer's comments below. As you will see and as you expected, this reviewer's opinion is much more positive, and we therefore agree with you that further consideration of your work is warranted. I would thus like to invite you to upload a final version of your work for publication here that addresses the following reviewer and editorial points:

- Please address the minor comment of reviewer #1
- Please link your profile in our submission system to your ORCID iD - you should have received an email with instructions on how to do so
- Please upload all figures, including supplementary figures, as individual files; all figure legends should remain in the main manuscript file
- Please provide your manuscript file in docx format
- The tables should remain in the main manuscript file (docx format)
- Figure S1 panels do not match figures S1 legend; please fix
- Figures S2: red data points not explained
- Please add scale bars where missing (e.g. Fig 1F, Figure 4D, Figure 6B,D, figure 7F)
- Please add description of arrows and arrow heads to the figure legends

Yours sincerely,

Reviewer #1:

The authors have considerably strengthened their manuscript by performing a significant amount of additional quantifications/experiments. In particular, the quantification/characterisation of VZ electroporated cells indicate that the vast majority of GFAP+ electroporated cells are non-B1 cells (probably due to their larger surface contacting the ventricle (i.e. and therefore exposed to electroporation) when compared to B1 cells). This, together with additional/parallel fate mapping experiments using the hGFAPCreERT2-Tom, greatly clarify the aim and conclusion of the manuscript (as illustrated in figures 1 and 9). All my concerns have been appropriately addressed. I therefore believe that this manuscript is suitable for publication in LSA

Minor comment:

The YFP+ cell presented in Figure 3C does not appear to be EdU positive. Please provide a better image.

May 18, 2020

RE: Life Science Alliance Manuscript #LSA-2020-00743-TRR-A

Dr. Karl J.L. Fernandes
University of Montreal
Neurosciences
900 rue St-Denis,
CRCHUM - Tour Viger, Neuroscience Dept, R09-482
Montreal, Quebec H2X 0A9
Canada

Dear Dr. Fernandes,

Thank you for submitting your Research Article entitled "Genetic targeting of neurogenic precursors in the adult forebrain ventricular epithelium". I appreciate the introduced changes and it is a pleasure to let you know that your manuscript is now accepted for publication in Life Science Alliance. Congratulations on this interesting work.

DISTRIBUTION OF MATERIALS:

Again, congratulations on a very nice paper. I hope you found the review process to be constructive and are pleased with how the manuscript was handled editorially. We look forward to future exciting

submissions from your lab.

Sincerely,
